# Policy Evaluation with Latent Confounders via Optimal Balance

**Andrew Bennett**[*]
Cornell University
awb222@cornell.edu

**Nathan Kallus**[*]
Cornell University
kallus@cornell.edu

## Abstract

Evaluating novel contextual bandit policies using logged data is crucial in applications where exploration is costly, such as medicine. But it usually relies on the assumption of no unobserved confounders, which is bound to fail in practice. We study the question of policy evaluation when we instead have proxies for the latent confounders and develop an importance weighting method that avoids fitting a latent outcome regression model. We show that unlike the unconfounded case no single set of weights can give unbiased evaluation for all outcome models, yet we propose a new algorithm that can still provably guarantee consistency by instead minimizing an adversarial balance objective. We further develop tractable algorithms for optimizing this objective and demonstrate empirically the power of our method when confounders are latent.

## 1 Introduction

Personalized intervention policies are of increasing importance in education [32], healthcare [3], and public policy [26]. In many of these domains exploration is costly or otherwise prohibitive, and so it is crucial to evaluate new policies using existing observational data. Usually, this relies on an assumption of no unobserved confounding (aka unconfoundedness or ignorability): that conditioned on observables, interventions are independent of idiosyncrasies that affect outcomes, so that counterfactuals can be reliably and correctly predicted. In particular, this enables the use of inverse propensity score (IPS) estimators of policy value [4, 21, 29] that eschew the need to actually fit outcome prediction models and doubly robust estimators that work even if such models are misspecified [8].

In practice, however, it may be unlikely that we observe confounders exactly. Nonetheless, if we observe very many features they may serve as good proxies for the true confounders, which can enable an alternative route to identification [22, 30]. In particular, noisy observations of true confounders can serve as valid proxies. For example, if intelligence is latent but affects both selection and outcome, we can instead use many noisy observations of intelligence such as school grades, IQ test, etc. Similarly, many medical measurements taken together can serve as proxies for underlying healthfulness.

In this paper, we study the problem of policy evaluation from observational data where we observe proxies instead of true confounders and we develop new weighting estimators based on optimizing balance in the latent confounders. Unlike the unconfounded setting where IPS weights ensure balance regardless of outcome model, we show that in this new setting there cannot exist any single of weights that ensure such unbiasedness regardless of outcome model. Instead, we develop an adversarial objective that bounds the conditional mean square error (CMSE) of any weighted estimator and, by appealing to game theoretic and empirical process arguments, we show that this objective can actually be driven to zero by a single set of weights. We therefore propose a novel policy evaluation

---

[*]Alphabetical order.

method that minimizes this objective, thus provably ensuring consistent estimation in the face of latent confounders. We develop tractable algorithms for this optimization problem. Finally, we provide empirical evidence demonstrating our method's consistent evaluation compared to standard evaluation methods and its improved performance compared to using fitted latent outcome models.

## 2 Problem

### 2.1 Setting and Assumptions

We consider a contextual decision making setting with $m$ possible treatments (aka actions or interventions). Each unit is associated with a set of potential outcomes $Y(1), \ldots, Y(m) \in \mathbb{R}$ corresponding to the reward/loss for each treatment, an observed treatment $T \in \{1, \ldots, m\}$, an observed outcome $Y = Y(T)$, true but latent confounders $Z \in \mathcal{Z} \subseteq \mathbb{R}^p$, and observed covariates $X \in \mathcal{X} \subseteq \mathbb{R}^q$. Our data consists of iid observations $X_i, T_i, Y_i$ of $X, T, Y$. Both the latent confounders and potential outcomes of unassigned treatments are unobserved. Note that $Y_i = Y_i(T_i)$ encapsulates the assumptions of consistency between observed and potential outcomes and non-interference between units.

A *policy* is a rule for assigning the probability of each treatment option given the observed covariates $X$. Given a policy $\pi$, we use the notation $\pi_t(x)$ to indicate the probability of assigning treatment $t$ when observed covariates are $x$. We define the *value* of a policy, $\tau^\pi$, as the expected outcome that would be obtained from following the policy in the population. Formally:

**Definition 1** (Policy Value). $\tau^\pi = \mathbb{E}[\sum_{t=1}^m \pi_t(X) Y(t)]$.

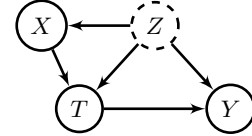

We encapsulate the assumption that $Z$ are sufficient for unconfoundedness and that $X$ is a proxy for $Z$ in the following assumption. Figure 1 provides a representation of this setting using a causal DAG [36]. Note importantly that we do *not* assume ignorability given $X$.

Figure 1: DAG representation of problem.

**Assumption 1** (Z are true confounders). *For every $t \in \{1, \ldots, m\}$, $Y(t)$ is independant of $(X, T)$, given $Z$.*

We next define the average mean outcome given $Z$ and its conditional expectations given observables:

$$\mu_t(z) = \mathbb{E}[Y(t) \mid Z = z],$$
$$\nu_t(x, t') = \mathbb{E}[\mu_t(Z) \mid X = x, T = t'] = \mathbb{E}[Y(t) \mid X = x, T = t'],$$
$$\rho_t(x) = \mathbb{E}[\mu_t(Z) \mid X = x] = \mathbb{E}[Y(t) \mid X = x].$$

We further define the propensity function and its conditional expectation given observables:

$$e_t(z) = \mathbb{P}(T = t \mid Z = z),$$
$$\eta_t(x) = P(T = t \mid X = x) = \mathbb{E}[e_t(Z) \mid X = x].$$

Finally, we denote by $\varphi(z; x, t)$ the conditional density of $Z$ given $X = x, T = t$. This density represents the latent variable model underlying the observables. For example, this can be a Gaussian mixture model, a PCA-type model as in [22], or a deep variational autoencoder as in [30]. Because we focus on how one might use such a latent model rather than the estimation of this model, we just assume we have some approximate oracle $\hat{\varphi}$ for calculating its values, such that $\hat{\varphi} = \varphi + O_p(1/\sqrt{n})$ in $\mathcal{L}^1$. (Note that for fair comparison, in experiments in Section 5, we similarly let the outcome regression methods use this oracle.)

We further make the following regularity assumptions:

**Assumption 2** (Weak Overlap). $\mathbb{E}\left[e_t^{-2}(Z)\right] < \infty$

**Remark 1.** *Given Assumption 2 it trivially follows that for every $t \in \{1, \ldots, m\}$, $x \in \mathcal{X}$, $z \in \mathcal{Z}$ that $e_t(z) > 0$ and $\eta_t(x) > 0$.*

**Assumption 3** (Bounded Variance). *The conditional variance of our potential outcomes given $X, T$ is bounded:* $\mathbb{V}[Y(t) \mid X, T] \leqslant \sigma^2$.

### 2.2 The Policy Evaluation Task

The problem we consider is to estimate the policy value $\tau^\pi$ given a policy $\pi$ and data $X_{1:n}, T_{1:n}, Y_{1:n}$. One standard approach to this is the *direct method* [38], which given an estimate $\hat{\rho}_t$ of $\rho_t$ predicts the

policy value as

$$\hat{\tau}_{\hat{\rho}}^{\pi} = \frac{1}{n} \sum_{i=1}^{n} \sum_{t=1}^{m} \pi_t(X_i) \hat{\rho}_t(X_i). \tag{1}$$

However this method is known to be biased and doesn't generalize well [4]. Furthermore given Assumption 1 $\hat{\rho}$ is not straightforward to estimate, since the mean value of $Y$ observed in our logged data given $X = x$ and $T = t$ is $\nu_t(x,t)$ not $\rho_t(x)$, so fitting $\hat{\rho}$ would require controlling for the effects of the unobserved $Z$.

An alternative to this is to come up with weights $W_{1:n} = f_W(X_{1:n}, T_{1:n})$ according to some function $f_W$ of the observed covariates and treatments, in order to re-weight the outcomes to look more like those that would be observed under $\pi$. Using these weights we can define the *weighted* estimator

$$\hat{\tau}_W^{\pi} = \frac{1}{n} \sum_{i=1}^{n} W_i Y_i. \tag{2}$$

This weighted estimator has the advantage that it does not require modeling the outcome distributions. Furthermore we could combine the weights $W_{1:n}$ with an outcome model $\hat{\rho}_t$ to calculate the *doubly robust* estimator [8], which is defined as

$$\hat{\tau}_{W,\hat{\rho}}^{\pi} = \frac{1}{n} \sum_{i=1}^{n} \sum_{t=1}^{m} \pi_t(X_i) \hat{\rho}_t(X_i) + \frac{1}{n} \sum_{i=1}^{n} W_i(Y_i - \hat{\rho}_{T_i}(X_i)). \tag{3}$$

The doubly robust estimator is known to be consistent when either the weighted or direct estimator is consistent and can attain local efficiency [19, 39].

Various approaches exist for coming up with weights for either the weighted or doubly robust estimators, which we discuss below. However none of these methods are applicable given Assumption 1, and so we develop a theory for weighting using proxy variables in Section 3.

## 2.3 Related Work

One of the most standard approaches for policy evaluation is using the weighted or doubly robust estimator defined in Eqs. (2) and (3), using inverse propensity score (IPS) weights. These are given by $W_i = \pi_{T_i}(X_i)/e_{T_i}(Z_i)$ [5], where $e_t$ are known or estimated logging probabilities. Since these weights can be extreme, both normalization [2, 31, 41] and clipping [10, 12, 40] are often employed. In addition some other approaches include recursive partitioning [14]. None of these methods are applicable to our setting however, since we do not know the true confounders $Z_{1:n}$.

An alternative to approaches based on fixed formulae for computing the importance weights is to compute weights that optimize an imbalance objective function [1, 15, 17, 18]. For policy evaluation, Kallus [16] propose to choose weights that adversarially minimize the conditional mean squared error of policy evaluation in the worst case of possible mean outcome functions in some reproducing kernel Hilbert space (RKHS), by solving a linearly constraint quadratic program (LCQP). Our work follows a very similar style to this, however instead of using the true confounders we only assume access to proxies, and we prove our theory for more general families of functions. This, we show, provides a unique imperative – separate from stability and variance control – to obtain importance weights via optimal balancing: while no single unbiased importance weights exist when we have proxies instead of true confounders, optimal balancing obtains weights in a model-robust manner that ensures consistency.

Finally there has been a long history of work in causal inference using proxies for true confounders [11, 42]. As in our problem setup, much of this work is based on the model of using an identified latent variable model for the proxies [9, 27, 34, 37, 43]. Some recent work on this problem involves using techniques such as matrix completion [22] or variational autoencoders [30] to infer confounders from the proxies. Additionally, there is recent work on robustness to unidentifiability due to unobserved confounding [20, 23, 24]; in contrast we focus on a setting where, while confounders are unobserved, effects are identifiable via proxies. In particular, there is a variety of work that studies sufficient conditions for the identifiability of latent confounder models [6, 34, 37]. Our work is complementary to this line of research in that we assume access to an accurate latent confounder model, but do not study how to estimate such models. Furthermore our work is novel in combining proxy variable models with optimal balancing and applying it to finding importance weights for policy evaluation.

## 3  Weight-Balancing Objectives

### 3.1  Infeasibility of IPS-Style Unbiased Weighting

If we had unconfoundedness given $X$ (*i.e.*, $Y(t) \perp\!\!\!\perp T \mid X$), the IPS weights $\pi_T(X)/\eta_T(X)$ are immediately gotten as the solution to making every term in the weighted sum Eq. (2) unbiased:

$$\mathbb{E}[W(X,T)\delta_{T_i t}Y(t)] = \mathbb{E}[\pi_t(X)Y(t)]. \tag{4}$$

Notably the IPS weights do not depend on the outcome function. However, without unconfoundedness given $X$ and given only Assumptions 1 to 3, this approach fails.

**Theorem 1.** *If $W(x,t)$ satisfies Eq. (4) then for any $t \in \{1, \dots, m\}$*

$$W(X,t) = \pi_t(X)\frac{\sum_{t' \in \tau} \eta_{t'}(X)\nu_t(X,t') + \Omega_t(X)}{\eta_t(X)\nu_t(X,t)}, \tag{5}$$

*for some $\Omega_t(x)$ such that $\mathbb{E}\Omega_t(X) = 0 \ \forall t$.*

The proof of Theorem 1 is given in Appendix A.1.

Note that *if* we had unconfoundedness given $X$ then $\nu_t(x,t') = \nu_t(x,t) = \rho_t(x)$ so that choosing $\Omega_t(X) = 0$ would recover the standard IPS weights. However, in our setting we generally have $\nu_t(x,t') \neq \nu_t(x,t)$, and so Theorem 1 tells us that we cannot do unbiased IPS-style weighted evaluation without knowing the mean outcome functions $\nu_t(x,t')$. In particular, there exists no single weight function that is simultaneously unbiased for all outcome functions.

On the other hand, Theorem 1 tells us that there do exist some weights that give unbiased and consistent policy evaluation via Eq. (2) or Eq. (3): we just may not be able to calculate them. The existence of such weights motivates our subsequent approach, which seeks weights that mimic these weights for a wide class of possible outcome functions.

### 3.2  Adversarial Error Objective

Over all weights that are functions of $X_{1:n}, T_{1:n}$, the *optimal* choice of weights for estimating $\tau^\pi$ via Eq. (2) would minimize the (unknown) conditional MSE (CMSE):

$$\mathbb{E}[(\hat{\tau}_W^\pi - \tau^\pi)^2 \mid X_{1:n}, T_{1:n}]. \tag{6}$$

In particular, the weights in Eq. (5) achieve $O_p(1/n)$ control on this CMSE for many outcome functions, as long as the denominator is well behaved, which can be seen by applying concentration inequalities to Eq. (6). However, as discussed above the outcome function is unknown and these weights are therefore practically infeasible. Our aim is to find weights with similar near-optimal behavior but that do not depend on the particular unknown outcome function. To do this, we will find an upper bound for Eq. (6) that we can actually compute.

Let $f_{it} = W_i \delta_{T_i t} - \pi_t(X_i)$ and

$$J^*(W,\mu) = \left(\frac{1}{n}\sum_{i=1}^{n}\sum_{t=1}^{m} f_{it}\nu_t(X_i, T_i)\right)^2 + \frac{2\sigma^2}{n^2}\|W\|_2^2,$$

where we embedded the dependence on $\mu$ inside $\nu_t(x,t') = \mathbb{E}[\mu_t(Z) \mid X = x, T = t']$.

**Theorem 2.** $\mathbb{E}[(\hat{\tau}_W^\pi - \tau^\pi)^2 \mid X_{1:n}, T_{1:n}] \leqslant 2J^*(W,\mu) + O_p(1/n).$

Note that $J^*$ above is defined in terms of the true posterior $\varphi$, which is infeasible to compute in practice. Therefore we define $J$ by replacing the conditional measure of $Z$ given $X$ and $T$ used to compute $\nu_t$ in $J^*$ with the approximate measure given by $\hat{\varphi}$. Note that in the remainder of this section and corresponding proofs we make slight abuse of notation by defining $J$ in terms of $\nu_t$; these $\nu_t$ terms should be interpreted below as defined in terms of the approximate conditional measure given by $\hat{\varphi}$. Then applying the fact that $\hat{\varphi} = \varphi + O_p(1/\sqrt{n})$ in $\mathcal{L}^1$, we obtain the following corollary which trivially follows from Theorem 2 and Slutsky's theorem:

**Corollary 1.** $\mathbb{E}[(\hat{\tau}_W^\pi - \tau^\pi)^2 \mid X_{1:n}, T_{1:n}] \leqslant 2J(W,\mu) + O_p(1/\sqrt{n})\sqrt{J(W,\mu)} + O_p(1/n).$

Therefore, if we find weights that obtain $O_p(1/n)$ control on $J(W, \mu)$, we can ensure that we also have $O_p(1/n)$ control on $\mathbb{E}[(\hat{\tau}_W^\pi - \tau^\pi)^2 \mid X_{1:n}, T_{1:n}]$. Combined with the following result, which follows from [13, Lemma 31], this would give root-n consistent estimation.

**Lemma 1.** *If* $\mathbb{E}[(\hat{\tau}_W^\pi - \tau^\pi)^2 \mid X_{1:n}, T_{1:n}] = O_p(1/n)$ *then* $\hat{\tau}_W^\pi = \tau^\pi + O_p(1/\sqrt{n})$.

It remains to find weights that control $J(W, \mu)$. The key obstacle for this is that $\mu$ is unknown. Instead, we show how we can obtain weights that control $J(W, \mu)$ over a whole class of given functions $\mu$.

Suppose we are given a set $\mathcal{F}$ of functions mapping $\mathcal{Z}$ to $\mathbb{R}^m$, where each $\mu \in \mathcal{F}$ corresponds to a vector of mean outcome functions $\mu = (\mu_1, \ldots, \mu_m)$. Then, motivated by Theorem 2 and Lemma 1, we define our adversarial optimization problem as

$$\mathcal{W}^* = \arg\min_{W \in \mathcal{W}} \sup_{\mu \in \mathcal{F}} J(W, \mu). \tag{7}$$

One question the reader might ask at this point is why not solve the above optimization problem by ignoring the hidden confounders and directly balancing the conditional mean outcome functions $\nu_t(x, t')$? The problem is that this would be impossible to do over any kind of generic flexible function space, since we have no data corresponding to terms in the form $\nu_t(x, t')$ when $t \ne t'$, so this is akin to an overlap problem. Conversely, if we were to ignore the conditioning on $t$ and balance against functions of the form $\nu_t(x) = \nu_t(x, t)$ this would be inadequate, as we couldn't hope for such a space to cover the true $\mu$ since we don't assume ignorability given $Z$.

In light of these limitations, we can view what we are doing in optimizing Eq. (7), using an identified model $\hat{\varphi}(z; x, t)$, as implicitly balancing some controlled space of functions $\nu_t(x, t')$ that do not have this overlap issue between different $t$ values. The following lemma makes this explicit, as it implies that that the terms $\nu_t(x, t')$ are all mutually bounded by each other for fixed $x$ and $t$:

**Lemma 2.** *Assuming* $\|\mu_t\|_\infty \le b$, *under Assumption 2, for all* $x \in \mathcal{X}$, *and* $t, t', t'' \in \{1, \ldots, m\}$ *we have*

$$|\nu_t(x, t'')| \le \frac{\eta_{t'}(x)}{\eta_{t''}(x)} \sqrt{8b\mathbb{E}\left[e_t^{-2}(Z) \mid X = x, T = t'\right]} |\nu_t(x, t')|.$$

### 3.3 Consistency of Adversarially Optimal Estimator

Now we analyze the consistency of our weighted estimator based on Eq. (7). Given Lemma 1, all we need to justify to prove consistency is that $\mu \in \mathcal{F}$ and that $\inf_W \sup_{\mu \in \mathcal{F}} J(W, \mu) \in O_p(\frac{1}{n})$. Define $\mathcal{F}_t$ as the space of all functions for treatment level $t$ allowed by $\mathcal{F}$. That is $\mathcal{F}_t = \{\mu_t : \exists (\mu_1', \ldots, \mu_m') \in \mathcal{F}$ with $\mu_t' = \mu_t\}$. We will use the following assumptions about $\mathcal{F}$ to prove control of $J$:

**Assumption 4** (Normed). *For each* $t \in \{1, \ldots, m\}$ *there exists a norm* $\|\cdot\|_t$ *on* $\text{span}(\mathcal{F}_t)$, *and there exists a norm* $\|\cdot\|$ *on* $\text{span}(\mathcal{F})$ *which is defined given some* $\mathbb{R}^m$ *norm as* $\|\mu\| = \|(\|\mu_1\|_1, \ldots, \|\mu_m\|_m)\|$.

**Assumption 5** (Absolutely Star Shaped). *For every* $\mu \in \mathcal{F}$ *and* $|\lambda| \le 1$, *we have* $\lambda\mu \in \mathcal{F}$.

**Assumption 6** (Convex Compact). $\mathcal{F}$ *is convex and compact*

**Assumption 7** (Square Integrable). *For each* $t \in \{1, \ldots, m\}$ *the space* $\mathcal{F}_t$ *is a subset of* $\mathcal{L}^2(\mathcal{Z})$, *and its norm dominates the* $\mathcal{L}^2$ *norm (i.e.,* $\inf_{\mu_t \in \mathcal{F}_t} \|\mu_t\|/\|\mu_t\|_{\mathcal{L}^2} > 0$).

**Assumption 8** (Nondegeneracy). *Define* $\mathcal{B}(\gamma) = \{\mu \in \text{span}(\mathcal{F}) : \|\mu\| \le \gamma\}$. *Then we have* $\mathcal{B}(\gamma) \subseteq \mathcal{F}$ *for some* $\gamma > 0$.

**Assumption 9** (Boundedness). $\sup_{\mu \in \mathcal{F}} \|\mu\|_\infty < \infty$.

**Definition 2** (Rademacher Complexity). $\mathcal{R}_n(\mathcal{F}) = \mathbb{E}[\sup_{f \in \mathcal{F}} \frac{1}{n} \sum_{i=1}^n \epsilon_i f(Z_i)]$, *where* $\epsilon_i$ *are iid Rademacher random variables*.

**Assumption 10** (Complexity). *For each* $t \in \{1 \ldots, m\}$ *we have* $\mathcal{R}_n(\mathcal{F}_t) = o(1)$.

These assumptions are satisfied for many commonly-used families of functions, such as RKHSs and families of neural networks. We shall prove this claim for RKHSs in Section 4.

In order to justify that we can control $J$, first we will show that these assumptions allow us to reverse the order of minimization and maximization in our optimization problem. This means we can reduce the problem to finding weights to control any particular $\mu$ rather than controlling all of $\mathcal{F}$.

**Lemma 3.** *Let $B(W, \mu) = \frac{1}{n} \sum_{i=1}^{n} \sum_{t=1}^{m} f_{it} \nu_t(X_i, T_i)$. Then under Assumptions 5 to 7 for every $M > 0$ we have the bound*

$$\min_W \sup_{\mu \in \mathcal{F}} J(W, \mu) \leqslant \sup_{\mu \in \mathcal{F}} \min_{\|W\|_2 \leqslant M} B(W, \mu)^2 + \frac{\sigma^2}{n^2} M^2.$$

Next, we note that Lemma 3 means that we can choose weights given $\mu$ to set $B(W, \mu) = 0$, and therefore we have our desired control as long as we can justify that these weights have controlled euclidean norm. Using this strategy and optimizing for the minimum norm weights of this kind, we are able to prove the following:

**Lemma 4.** *Under Assumptions 4 to 10 we have $\inf_W \sup_{\mu \in \mathcal{F}} J(W, \mu) = O_p(1/n)$.*

This is the key lemma in proving our main consistency theorem:

**Theorem 3.** *Under Assumptions 4 to 10 and assuming that $\mu \in \mathcal{F}$ we have $\hat{\tau}_{W*}^{\pi} = \tau^{\pi} + O_p(1/\sqrt{n})$.*

This theorem follows immediately from our previous results, since $\mu \in \mathcal{F}$ and Lemma 4 imply that $J(W^*, \mu) = O_p(1/n)$. This combined with Corollary 1 imply that $\mathbb{E}[(\hat{\tau}_{W*}^{\pi} - \tau^{\pi})^2 \mid X_{1:n}, T_{1:n}] = O_p(1/n)$, which in turn combined with Lemma 1 gives us our result. Furthermore, given some additional assumptions, we can take advantage of $\mathcal{L}^{\infty}$ universal approximation of $\mu$ to obtain the following corollary which does *not* depend on the assumption that $\mu \in \mathcal{F}$:

**Assumption 11** (Continuous Mean Outcome). *For each $t$, $\mu_t$ is a continuous function of $Z$.*

**Assumption 12** (Universal Approximation). *$\mathcal{F}_m$ is a universally approximating sequence of function classes. That is, for every vector of continuous functions $\mu$ and every $m$, there exists $f \in \mathcal{F}_m$ such that $\|f_t - \mu_t\|_{\infty} \leqslant \epsilon_m$ for each $t$, and $\epsilon_m \to 0$.*

**Corollary 2.** *Under Assumptions 4 to 12 and using a universally approximating sequence of function classes $\mathcal{F}_n$ to compute $W^*$, we have $\hat{\tau}_{W*}^{\pi} = \tau^{\pi} + O_p(1)$.*

This corollary follows from observing that by the universally approximating property of $\mathcal{F}_n$, we can obtain an $O_p(1/\sqrt{n}) + \epsilon$ error bound for every $\epsilon > 0$ (where the $O_p$ term's constants can depend on $\epsilon$). Thus $O_p(1)$ error bound follows trivially. Note that the universal approximation property of Assumption 12 is obtainable for many classes of functions such as Gaussian RKHSs [35].

## 4   Algorithms for Optimal Kernel Balancing

### 4.1   Kernel Function Class

We now provide an algorithm for optimal balancing when our function class consists of vectors of RKHS functions. Formally, given a kernel $K$ and corresponding RKHS norm $\|\cdot\|_K$, we define the space $\mathcal{F}^K$ as follows:

**Definition 3** (Kernel Class). *$\mathcal{F}^K = \{\mu : \|\mu\| \leqslant 1\}$, where $\|(\mu_1, \ldots, \mu_m)\| = \sqrt{\sum_{t=1}^{m} \|\mu_t\|_K^2}$.*

**Theorem 4.** *Assuming $K$ is a Mercer kernel [44] and is bounded, $\mathcal{F}^K$ satisfies Assumptions 4 to 10.*

We can remark that the commonly used Gaussian kernel is both Mercer and bounded, so it satisfies the conditions of Theorem 4. Given this, and assuming that $\mathcal{F}^K$ covers the real mean outcome function $\mu$, we can apply Theorem 3 to see that solving Eq. (7) using $\mathcal{F}^K$ gives consistent evaluation.

Note that the $\mathcal{F}^K$ having maximum norm 1 is without loss of generality, because if we wanted the maximum norm to instead be $m$ we could replace the standard deviation $\sigma$ by $\sigma/m$ in our objective function, resulting in an equivalent re-scaled optimization problem. To make this explicit, we will replace $\sigma$ in the objective with $\gamma$, where it is assumed that $\gamma$ is a freely chosen hyperparameter to allow for varying regularization. For ease of notation below we define $\Gamma$ to be the diagonal matrix such that $\Gamma_{ii} = \gamma$ for every $i$.

### 4.2   Kernel Balancing Algorithm

In order to optimize Eq. (7) over a class of kernel functions as defined by Definition 3, we can observe that the definition of $J(W, \mu)$ looks very similar to the adversarial objective of Kallus [16],

except that we have $\nu_t(X_i, T_i)$ terms instead of $\mu_t(X_i)$ terms. This motivates the idea that, given our identified posterior model $\hat{\varphi}(z; x, t)$, we may be able to employ a similar quadratic programming (QP)-based approach. The following theorem makes this explicit, by defining a QP objective for $W$ that we can approximate by sampling from $\hat{\varphi}$:

**Theorem 5.** *Define* $Q_{ij} = \mathbb{E}[K(Z_i, Z_j')]$, $G_{ij} = \frac{1}{n^2}(Q_{ij}\delta_{T_iT_j} + \Gamma_{ij})$, *and* $a_i = \frac{2}{n^2}\sum_{j=1}^n Q_{ij}\pi_{T_j}(X_i)$, *where for each $i$ $Z_i$ and $Z_i'$ are iid shadow variables, and the expectation is defined condional on the observed data using the approximate posterior $\hat{\varphi}$. Then for some $c$ that is constant in $W$ we have the identity*

$$\sup_{\mu \in \mathcal{F}^K} J(W, \mu) = W^T G W - a^T W + c.$$

Given this our balancing algorithm is natural and straightforward, and is summarized by Algorithm 1. Note that we provide an optional weight space constraint $\mathcal{W}$ in this algorithm, since standard weighted estimator approaches for policy evaluation regularize by forcing constraints such as $W \in n\Delta^n$. Under this kind of constraint our unconstrained QP becomes a LCQP. However our theory does not support this constraint, and we find that it hurts performance in practice, especially when $\Gamma$ is large, so we do not use this constraint in our main experiments.

---

**Algorithm 1** Optimal Kernel Balancing

---

**Input:** Data $(X_{1:n}, T_{1:n})$, policy $\pi$, kernel function $K$, posterior density $\hat{\varphi}$, regularization matrix
$\quad\quad$ $\Gamma$, number samples $B$, optional weight space $\mathcal{W}$ (defaults to $\mathbb{R}^n$ if not provided)
**Output:** Optimal balancing weights $W_{1:n}$
1: **for** $i \in \{1, \ldots, n\}$ **do**
2: $\quad$ **Sample Data.** Draw $B$ data points $Z_i^b$ from the posterior $\hat{\varphi}(\cdot\ ; X_i, T_i)$
3: **end for**
4: **Estimate Q.** Calculate $Q_{ij} = \frac{1}{B^2}\sum_{b=1}^B\sum_{c=1}^B K(Z_i^b, Z_i^c)$
5: **Calculate QP Inputs.** Calculate $G_{ij} = Q_{ij}\delta_{T_iT_j} + \Gamma_{ij}$, and $a_i = 2\sum_{j=1}^n Q_{ij}\pi_{T_j}(X_i)$,
6: **Solve Quadratic Program.** Calculate $W = \arg\min_{W\in\mathcal{W}} W^T G W - a^T W$

---

## 5 Experiments

### 5.1 Experimental Setup

We now present a brief set of experiments to explore our methodology. The aim of these experiments is to be a proof of concept of our theory. We seek to show that given an identified posterior model $\hat{\varphi}$ policy as discussed in Section 2.1, evaluation using the weights defined by Eq. (7) can give unbiased policy evaluation even in the face of sufficiently strong confounding where standard benchmark approaches that rely on ignorability given $X$ fail. We experiment with the following generalized linear model-style scenario:

$$
\begin{array}{lll}
Z \sim \mathcal{N}(0,1) & X \sim \mathcal{N}(\alpha^T Z + \alpha_0, \sigma_X) & P_T = \beta^T Z + \beta_0 \\
T \sim \text{softmax}(P_T) & W(t) \sim \mathcal{N}(\zeta(t)^T Z + \zeta_0(t), \sigma_Y) & Y(t) = g(W(t))
\end{array}
$$

In our experiments $Z$ is 1-dimensional, $X$ is 10-dimensional, and we have two possible treatment levels ($m = 2$). We experiment with a parametric policy $\pi$ and multiple link functions $g$ as follows:

$$\pi_t(X) = \frac{\exp(\psi_t^T X)}{\exp(\psi_1^T X) + \exp(\psi_2^T X)}$$

**step**: $g(w) = 3\mathbb{1}_{\{w\geq 0\}} - 6$ $\quad$ **exp**: $g(w) = \exp(w)$ $\quad$ **cubic**: $g(w) = w^3$ $\quad$ **linear**: $g(w) = w$

We experiment with the following methods in this evaluation:

1. **OptZ** Our method, using $\Gamma = \gamma \, \text{Identity}(n)$ for $\gamma \in \{0.001, 0.2, 1.0, 5.0\}$.
2. **IPS** IPS weights based on $X$ using estimated $\hat{\eta}_t$.
3. **OptX** The optimal weighting method of Kallus [16] with same values of $\Gamma$ as our method.

4. **DirX** Direct method by fitting $\hat{\rho}_t(x)$ incorrectly assuming ignorability given $X$.

5. **DirZ** Direct method by first fitting $\hat{\mu}_t$ using posterior samples from $\hat{\varphi}$, then using the estimate $\hat{\rho}_t(x) = (1/D) \sum_{i=1}^{D} \hat{\mu}_t(z_i')$, where $z_i'$ are sampled from $\hat{\varphi}(\cdot; x, t)$.

6. **D:W** Doubly robust estimation using direct estimator **D** and weighted estimator **W**.

Finally we detail all choices for scenario parameters in Appendix B.1, and provide implementation details of our methods in Appendix B.2.[2]

## 5.2 Results

We display results for our experiments using the **step** link function in Tables 1 and 2. For each of $n \in \{200, 500, 1000, 2000\}$ we estimate the RMSE of policy evaluation using each method, as well as doubly robust evaluation using our best performing weights, by averaging over 64 runs. In addition, in Tables 3 and 4 we display the estimated bias from the evaluations. It is clear that the naive methods that assume ignorability given $X$ all hit a performance ceiling, where bias converges to some non-zero value. In particular for IPS we separately ran it on up to one million data points and found that the bias converged to $0.418 \pm 0.001$. On the other hand, for our method it appears like we have consistency. This is particularly evident when we look at Table 3, as bias seems to be approximately converging to zero with vanishing variance. We can also observe that doubly robust estimation using either direct method does not appear to improve performance.

It is noteworthy that the **DirZ** benchmark method fails horribly, despite being a correctly specified regression estimate. From our experience we observed that it is difficult to train the $\mu_t$ functions accurately if there is a high amount of overlap in the $\hat{\varphi}(\cdot; x, t)$ posteriors for fixed $t$. Therefore we postulate that in highly confounded settings this benchmark is inherently difficult to train using a finite number of samples from $\hat{\varphi}(\cdot; x, t)$, and the result seems to collapse to degenerate solutions.

Next we note that we observed similar trends to this using our other link functions, and other doubly robust estimators. We present more extensive tables of results in Appendix C.1. In addition we present some results there on the negative impact on our method's performance using the constraint $W \in n\Delta^n$, as mentioned in Section 4.2.

Finally we provide some more detailed experiments investigating the impact of changing the dimensionality of $Z$ and the level of confounding by replacing $Z$ with $X$ in Appendix C.2 and Appendix C.3 respectively. In brief the results of these experiments are as expected: increasing the dimensionality of $Z$ gives the same overall pattern of results but with slower convergence, and increasing the level of confounding strongly decreases the performance of benchmark methods, while our method appears to maintain its unbiasedness.

# 6 Conclusion

We presented theory for how to do optimal balancing for policy evaluation when we only have proxies for the true confounders, given an already identified model for the confounders, treatment, and proxies, but not for the outcomes. We provided an adversarial objective for selecting optimal weights given some class of mean outcome functions to be balanced, and proved that under mild conditions these optimal weights result in consistent policy evaluation. In addition, we presented a tractable algorithm for minimizing this objective when our function class is an RKHS, and we conducted a series of experiments to demonstrate that our method can achieve consistent evaluation even under sufficient levels of confounding where standard approaches fail.

For future work we note that the adversarial objective and theory presented here is fairly general, and could be used to develop new algorithms for balancing different function classes such as neural networks. Indeed neural networks with weight decay easily satisfy the conditions of both Theorem 3 and Corollary 2, and thus it might be possible to learn balancing weights by optimizing a GAN-like objective. An alternative direction would be to study how best to apply this methodology when an identified model is not already given.

| n | $\text{OptZ}_{0.001}$ | $\text{OptZ}_{0.2}$ | $\text{OptZ}_{1.0}$ | $\text{OptZ}_{5.0}$ | $\text{DirX:OptZ}_{0.001}$ | $\text{DirZ:OptZ}_{0.001}$ |
|---|---|---|---|---|---|---|
| 200 | $.39 \pm .07$ | $.24 \pm .02$ | $.36 \pm .02$ | $.81 \pm .02$ | $.57 \pm .06$ | $.41 \pm .07$ |
| 500 | $.19 \pm .02$ | $.18 \pm .02$ | $.23 \pm .02$ | $.49 \pm .02$ | $.55 \pm .03$ | $.20 \pm .02$ |
| 1000 | $.11 \pm .01$ | $.11 \pm .01$ | $.13 \pm .01$ | $.27 \pm .01$ | $.49 \pm .02$ | $.11 \pm .01$ |
| 2000 | $.08 \pm .01$ | $.08 \pm .01$ | $.09 \pm .01$ | $.17 \pm .01$ | $.48 \pm .01$ | $.08 \pm .01$ |

Table 1: Convergence of RMSE for for policy evaluation using our weights.

| n | IPS | $\text{OptX}_{0.001}$ | $\text{OptX}_{0.2}$ | $\text{OptX}_{1.0}$ | $\text{OptX}_{5.0}$ | DirX | DirZ |
|---|---|---|---|---|---|---|---|
| 200 | $.47 \pm .03$ | $2.0 \pm .03$ | $2.1 \pm .03$ | $2.3 \pm .02$ | $2.5 \pm .02$ | $.52 \pm .02$ | $2.6 \pm .02$ |
| 500 | $.48 \pm .03$ | $2.0 \pm .02$ | $2.1 \pm .02$ | $2.3 \pm .02$ | $2.6 \pm .02$ | $.48 \pm .02$ | $2.6 \pm .01$ |
| 1000 | $.39 \pm .02$ | $2.0 \pm .01$ | $2.1 \pm .01$ | $2.3 \pm .01$ | $2.5 \pm .01$ | $.48 \pm .02$ | $2.6 \pm .01$ |
| 2000 | $.40 \pm .01$ | $2.0 \pm .01$ | $2.1 \pm .01$ | $2.3 \pm .01$ | $2.5 \pm .01$ | $.45 \pm .02$ | $2.6 \pm .01$ |

Table 2: Convergence of RMSE for benchmark methods.

| n | $\text{OptZ}_{0.001}$ | $\text{OptZ}_{0.2}$ | $\text{OptZ}_{1.0}$ | $\text{OptZ}_{5.0}$ | $\text{DirX:OptZ}_{0.001}$ | $\text{DirZ:OptZ}_{0.001}$ |
|---|---|---|---|---|---|---|
| 200 | $.03 \pm .39$ | $.11 \pm .21$ | $.29 \pm .21$ | $.78 \pm .18$ | $.43 \pm .38$ | $.05 \pm .40$ |
| 500 | $.09 \pm .17$ | $.10 \pm .15$ | $.17 \pm .16$ | $.47 \pm .15$ | $.51 \pm .19$ | $.10 \pm .18$ |
| 1000 | $.02 \pm .11$ | $.05 \pm .09$ | $.08 \pm .09$ | $.25 \pm .09$ | $.47 \pm .13$ | $.04 \pm .11$ |
| 2000 | $.03 \pm .07$ | $.05 \pm .06$ | $.07 \pm .07$ | $.16 \pm .07$ | $.47 \pm .09$ | $.03 \pm .07$ |

Table 3: Convergence of bias for policy evaluation using our weights.

| n | IPS | $\text{OptX}_{0.001}$ | $\text{OptX}_{0.2}$ | $\text{OptX}_{1.0}$ | $\text{OptX}_{5.0}$ | DirX | DirZ |
|---|---|---|---|---|---|---|---|
| 200 | $.40 \pm .25$ | $1.9 \pm .21$ | $2.1 \pm .20$ | $2.3 \pm .19$ | $2.5 \pm .18$ | $.49 \pm .18$ | $2.6 \pm .14$ |
| 500 | $.43 \pm .21$ | $2.0 \pm .16$ | $2.1 \pm .15$ | $2.3 \pm .14$ | $2.6 \pm .13$ | $.45 \pm .16$ | $2.6 \pm .12$ |
| 1000 | $.37 \pm .12$ | $2.0 \pm .10$ | $2.1 \pm .09$ | $2.3 \pm .09$ | $2.5 \pm .08$ | $.46 \pm .15$ | $2.6 \pm .11$ |
| 2000 | $.39 \pm .10$ | $2.0 \pm .08$ | $2.1 \pm .07$ | $2.3 \pm .07$ | $2.5 \pm .07$ | $.42 \pm .17$ | $2.6 \pm .11$ |

Table 4: Convergence of bias for benchmark methods.

## Acknowledgements

This material is based upon work supported by the National Science Foundation under Grant No. 1846210. This research was funded in part by JPMorgan Chase & Co. Any views or opinions expressed herein are solely those of the authors listed, and may differ from the views and opinions expressed by JPMorgan Chase & Co. or its affiliates. This material is not a product of the Research Department of J.P. Morgan Securities LLC. This material should not be construed as an individual recommendation for any particular client and is not intended as a recommendation of particular securities, financial instruments or strategies for a particular client. This material does not constitute a solicitation or offer in any jurisdiction.

## Footnotes

[2]Code available online at `https://github.com/CausalML/LatentConfounderBalancing`.

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
