[Supplementary Material]

# A Omitted Proofs

## A.1 Proof of Theorem 1

First we note that $W(X,T)Y - \sum_t \pi_t(X)Y(t)] = \sum_t (W(X,t)\delta_{Tt} - \pi_t(X))Y(t)$. Then analyzing each summand separately, we can obtain:

$$
\begin{aligned}
\mathbb{E}[W(X,t)&\delta_{Tt}Y(t) - \pi_t(X)Y(t)] \\
&= \mathbb{E}[\mathbb{E}[W(X,t)\delta_{Tt}Y(t) - \pi_t(X)Y(t) \mid X]] \\
&= \mathbb{E}[W(X,t)\mathbb{E}[\delta_{Tt}Y(t) \mid X] - \pi_t(X)\mathbb{E}[Y(t) \mid X]] \\
&= \mathbb{E}[W(X,t)\mathbb{E}[\mathbb{E}[\delta_{Tt}Y(t) \mid X,T] \mid X] - \pi_t(X)\mathbb{E}[\mathbb{E}[Y(t) \mid X,Z] \mid X]] \\
&= \mathbb{E}[W(X,t)\mathbb{E}[\delta_{Tt}\mathbb{E}[Y(t) \mid X,T=t] \mid X] - \pi_t(X)\mathbb{E}[\mu_t(Z) \mid X]] \\
&= \mathbb{E}[W(X,t)\mathbb{E}[\delta_{Tt} \mid X]\mathbb{E}[\mu_t(Z) \mid X,T=t] - \pi_t(X)\mathbb{E}[\mu_t(Z) \mid X]] \\
&= \mathbb{E}[W(X,t)\eta_t(X)\mathbb{E}[\mu_t(Z) \mid X,T=t] - \pi_t(X)\mathbb{E}[\mu_t(Z) \mid X]] \\
&= \mathbb{E}\left[W(X,t)\eta_t(X)\nu_t(X,t) - \pi_t(X)\sum_{t'}\eta_{t'}(X)\nu_t(X,t')\right]
\end{aligned}
$$

Therefore the solution to the equation $\mathbb{E}[W(X,t)\delta_{Tt}Y(t) - \pi_t(X)Y(t)] = 0$ is given by:

$$
W(X,t)\eta_t(X)\nu_t(X,t) - \pi_t(X)\sum_{t'}\eta_{t'}(X)\nu_t(X,t') = \Omega_t(X)
$$

where $\Omega_t(X)$ is any arbitrary function of $X$ with mean zero. Solving this for $W$ gives

$$
W(X,t) = \frac{\pi_t(X)\sum_{t'}\eta_{t'}(X)\nu_t(X,t') + \Omega_t(X)}{\eta_t(X)\nu_t(X,t)},
$$

and finally replacing $X$ with $x$ gives the required solution.

## A.2 Proof of Theorem 2

Define

$$
\tau^\pi_{SAPE} = \frac{1}{n}\sum_i \sum_t \pi_t(X_i)\mu_t(Z_i).
$$

Using $(x+y)^2 \leqslant 2x^2 + 2y^2$, we have

$$
\mathbb{E}[(\hat{\tau}^\pi_W - \tau^\pi)^2 \mid X_{1:n}, T_{1:n}] \leqslant 2\mathbb{E}[(\hat{\tau}^\pi_W - \tau^\pi_{SAPE})^2 \mid X_{1:n}, T_{1:n}] + 2\mathbb{E}[(\tau^\pi_{SAPE} - \tau^\pi)^2 \mid X_{1:n}, T_{1:n}].
$$

Noting that $\tau^\pi = \mathbb{E}[\tau^\pi_{SAPE}]$, Assumption 3 implies that $\mathbb{E}[(\tau^\pi_{SAPE} - \tau^\pi)^2] = O(1/n)$. Markov's inequality yields $\mathbb{E}[(\tau^\pi_{SAPE} - \tau^\pi)^2 \mid X_{1:n}, T_{1:n}] = O_p(1/n)$.

Let $CMSE(W,\mu) = \mathbb{E}[(\hat{\tau}^\pi_W - \tau^\pi_{SAPE})^2 \mid X_{1:n}, T_{1:n}]$. We proceed to bound $CMSE$. By iterating expectations we can obtain:

$$
\begin{aligned}
CMSE(W,\mu) &= \mathbb{E}[(\hat{\tau}^\pi_W - \tau^\pi_{SAPE})^2 \mid X,T] \\
&= \mathbb{E}[\mathbb{E}[(\hat{\tau}^\pi_W - \tau^\pi_{SAPE})^2 \mid X,T,Z \mid X,T] \\
&= \mathbb{E}[\mathbb{E}[\hat{\tau}^\pi_W - \tau^\pi_{SAPE} \mid X,T,Z]^2 \mid X,T] + \mathbb{E}[\mathbb{V}[\hat{\tau}^\pi_W - \tau^\pi_{SAPE} \mid X,T,Z] \mid X,T] \\
&= \mathbb{E}[(\frac{1}{n}\sum_{i,t} f_{it}\mu_t(Z_i))^2 \mid X,T] + \mathbb{E}[\mathbb{V}[\frac{1}{n}\sum_i W_i Y(T_i) \mid X,T,Z] \mid X,T] \\
&\leqslant \mathbb{E}[(\frac{1}{n}\sum_{i,t} f_{it}\mu_t(Z_i))^2 \mid X,T] + \frac{\sigma^2}{n^2}\|W\|_2^2
\end{aligned}
$$

where $\sigma$ is the bound defined in Assumption 3.

Next observe that for any (possibly correlated) random variables $A_1, \ldots, A_n$ and numbers $p_1, \ldots, p_n \in \mathbb{R}^n$ such that $\sum_i p_i = 1$, we have $\mathbb{V}[\sum_i p_i A_i] \leqslant \max_i \mathbb{V}[A_i]$. Given this, we can

simplify the first term above further, as follows:

$$\mathbb{E}[(\frac{1}{n}\sum_{i,t} f_{it}\mu_t(Z_i))^2 \mid X, T] = (\frac{1}{n}\sum_{i,t} f_{it}\mathbb{E}[\mu_t(Z_i) \mid X_i, T_i])^2 + \mathbb{V}[\frac{1}{n}\sum_{i,t} f_{it}\mu_t(Z_i) \mid X, T]$$

$$= (\frac{1}{n}\sum_{i,t} f_{it}\nu_t(X_i, T_i))^2 + \frac{1}{n^2}\sum_i \mathbb{V}[\sum_t f_{it}\mu_t(Z_i) \mid X, T]$$

$$\leqslant (\frac{1}{n}\sum_{i,t} f_{it}\nu_t(X_i, T_i))^2 + \frac{1}{n^2}\sum_i \max_t f_{it}^2 \mathbb{V}[\mu_t(Z_i) \mid X, T]$$

$$\leqslant (\frac{1}{n}\sum_{i,t} f_{it}\nu_t(X_i, T_i))^2 + \frac{2\sigma^2}{n^2}\sum_i \max_t W_i^2 \delta_{T_i t} + \pi_t(X_i)^2$$

$$\leqslant (\frac{1}{n}\sum_{i,t} f_{it}\nu_t(X_i, T_i))^2 + \frac{2\sigma^2}{n^2}\sum_i (W_i^2 + 1)$$

$$= (\frac{1}{n}\sum_{i,t} f_{it}\nu_t(X_i, T_i))^2 + \frac{2\sigma^2}{n^2}\|W\|_2^2 + \frac{2\sigma^2}{n}$$

where for the second inequality we used the fact that $(x + y)^2 \leqslant 2x^2 + 2y^2$. This gives us $CMSE(W, \mu) \leqslant J^*(W, \mu) + O_p(1/n)$, and combining this with the above gives $\mathbb{E}[(\hat{\tau}_W^\pi - \tau^\pi)^2 \mid X_{1:N}, T_{1:N}] \leqslant 2J^*(W, \mu) + O_p(1/n)$ as required.

## A.3   Proof of Lemma 2

First, we will use the notation $f(z; x, t)$ for the conditional measure of $Z$ given $X = x$ and $T = t$, and observe that according to Bayes rule we have:

$$\frac{f(z; x, t'')}{f(z; x, t')} = \frac{e_{t''}(z)}{e_{t'}(z)} \frac{\eta_{t'}(x)}{\eta_{t''}(x)}$$

Define $\mathbb{E}_{xt}$ and $\mathbb{P}_{xt}$ as shorthand for expectation and probability given $X = x, T = t$ respectively. Then given the above, for any $M > 0$ we can bound

$$\nu_t(x, t'') = \frac{\eta_{t''}(x)}{\eta_{t'}(x)}\mathbb{E}[\mu_t(Z) \mid X = x, T = t'']$$

$$= \int_{\mathcal{Z}} f(z; x, t'')\mu_t(z)dz$$

$$= \frac{\eta_{t''}(x)}{\eta_{t'}(x)} \int_{\mathcal{Z}} \frac{e_{t''}(z)}{e_{t'}(z)} \frac{\eta_{t'}(x)}{\eta_{t''}(x)} f(z; x, t')\mu_t(z)dz$$

$$= \frac{\eta_{t'}(x)}{\eta_{t''}(x)} \int_{\mathcal{Z}} \frac{e_{t''}(z)}{e_{t'}(z)} f(z; x, t')\mu_t(z)dz$$

$$\leqslant \frac{\eta_{t'}(x)}{\eta_{t''}(x)} \left( M\mathbb{E}_{xt'}[\mathbb{1}\{\frac{e_{t''}(z)}{e_{t'}(z)} \leqslant M\}\mu_t(Z)] + \mathbb{E}_{xt'}[\mathbb{1}\{\frac{e_{t''}(z)}{e_{t'}(z)} > M\}\frac{e_{t''}(z)}{e_{t'}(z)}\mu_t(Z)] \right)$$

Now we can use the fact that $\mu_t$ is $b$-bounded to bound the first term by

$$M\mathbb{E}_{xt'}[\mathbb{1}\{\frac{e_{t''}(z)}{e_{t'}(z)} \leqslant M\}\mu_t(Z)] = M\nu_t(x, t') - M\mathbb{E}_{xt'}[\mathbb{1}\{\frac{e_{t''}(z)}{e_{t'}(z)} > M\}\mu_t(Z)]$$

$$\leqslant M\nu_t(x, t') + Mb\mathbb{P}_{xt'}[\frac{e_{t''}(z)}{e_{t'}(z)} > M],$$

and in addition applying Cauchy Schwartz we can bound the second term by

$$\mathbb{E}_{xt'}[\mathbb{1}\{\frac{e_{t''}(z)}{e_{t'}(z)} > M\}\frac{e_{t''}(z)}{e_{t'}(z)}\mu_t(Z)] \leqslant \sqrt{\mathbb{E}_{xt'}[\mathbb{1}\{\frac{e_{t''}(z)}{e_{t'}(z)} > M\}\mu_t(Z)^2]\mathbb{E}_{xt'}[\left(\frac{e_{t''}(z)}{e_{t'}(z)}\right)^2]}$$

$$\leqslant \sqrt{b^2\mathbb{P}_{xt'}[\frac{e_{t''}(z)}{e_{t'}(z)} > M]\mathbb{E}_{xt'}[\left(\frac{e_{t''}(z)}{e_{t'}(z)}\right)^2]}$$

$$\leqslant b\sqrt{\mathbb{P}_{xt'}[\frac{1}{e_{t'}(z)} > M]\mathbb{E}_{xt'}[\left(\frac{1}{e_{t'}(z)}\right)^2]}.$$

Now define $g(x,t) = \mathbb{E}_{xt}[\left(\frac{1}{e_t(z)}\right)^2]$. By Assumption 2 we know $g(x,t)$ is finite for every $x$ and $t$. Also by Markov's inequality we know that $\mathbb{P}_{xt}[\frac{1}{e_t(z)} > M] \leqslant \frac{g(x,t)}{M^2}$. Therefore putting all of the above together we can obtain

$$\nu_t(x,t'') \leqslant \frac{\eta_{t'}(x)}{\eta_{t''}(x)}\left(M\nu_t(x,t') + \frac{2bg(x,t')}{M}\right)$$

$$\leqslant \frac{\eta_{t'}(x)}{\eta_{t''}(x)}\left(M|\nu_t(x,t')| + \frac{2bg(x,t')}{M}\right)$$

This inequality is valid every $M$, so we can pick $M$ to make it as tight as possible. Choosing $M = \sqrt{\frac{2bg(x,t')}{\nu_t(x,t')}}$ gives us:

$$\nu_t(x,t'') \leqslant \frac{\eta_{t'}(x)}{\eta_{t''}(x)}\sqrt{8bg(x,t')|\nu_t(x,t')|}$$

Finally note that, since by symmetry $\mathbb{E}[\mu(Z) \mid X,T] = -\mathbb{E}[-\mu(Z) \mid X,T]$, we can strengthen this inequality to the following

$$|\nu_t(x,t'')| \leqslant \frac{\eta_{t'}(x)}{\eta_{t''}(x)}\sqrt{8bg(x,t')|\nu_t(x,t')|},$$

and noting that $g(x,t') = \mathbb{E}[e_t^{-2}(Z) \mid X = x, T = t']$ gives us our final result.

### A.4   Proof of Lemma 3

First note that by Assumption 6 $\mathcal{F}$ is compact. Also $J(W, \cdot)$ is continuous for every $W$, since by Assumption 7 we know that the norm on each $\mathcal{F}_t$ dominates the norm on $\mathcal{L}^2(\mathcal{Z})$ and this continuity result would be trivial if $\mathcal{F}_t = \mathcal{L}^2(\mathcal{Z})$. This means that by the Extreme Value theorem we can replace the supremum over $\mu$ with a maximum over $\mu$ in the quantity we are bounding. Given this, we will proceed by bounding $\min_W \max_{\mu \in \mathcal{F}} B(W, \mu)$ using von Neumann's minimax theorem to swap the minimum and the maximum, and then use this to establish the overall bound for $J(W, \mu)$.

Next we can observe that $B(W, \mu)$ is linear, and therefore both convex and concave, for each of $W$ and $\mu$. Next, by Assumption 6 $\mathcal{F}$ is convex and compact, and following the same argument as above $B(W, \cdot)$ is continuous for every $W$. In addition, $B(W, \mu)$ is also clearly continuous in $W$ for fixed $\mu$, and the set $\{W : \|W\|_2 \leqslant M\}$ is obviously compact and convex for any constant $M$. Thus by von Neumann's minimax theorem we have the following for every finite $M$:

$$\min_{\|W\|_2 \leqslant M} \max_{\mu \in \mathcal{F}} B(W, \mu) = \max_{\mu \in \mathcal{F}} \min_{\|W\|_2 \leqslant M} B(W, \mu) \tag{8}$$

Given this, we can bound $\min_W \max_{\mu \in \mathcal{F}} \hat{J}(W, \mu)$ as follows, which is valid for any $M$:

$$\min_W \max_{\mu \in \mathcal{F}} \hat{J}(W, \mu) \leqslant \min_W \max_{\mu \in \mathcal{F}} B(W, \mu)^2 + \frac{1}{n^2} W^T \Sigma W$$

$$\leqslant \min_W \max_{\mu \in \mathcal{F}} B(W, \mu)^2 + \frac{\sigma^2}{n^2} \|W\|_2^2$$

$$\leqslant \min_{\|W\|_2 \leqslant M} \max_{\mu \in \mathcal{F}} B(W, \mu)^2 + \frac{\sigma^2}{n^2} \|W\|_2^2$$

$$\leqslant (\min_{\|W\|_2 \leqslant M} \max_{\mu \in \mathcal{F}} B(W, \mu))^2 + \frac{\sigma^2}{n^2} M^2$$

$$= (\max_{\mu \in \mathcal{F}} \min_{\|W\|_2 \leqslant M} B(W, \mu))^2 + \frac{\sigma^2}{n^2} M^2$$

$$= \max_{\mu \in \mathcal{F}} \min_{\|W\|_2 \leqslant M} B(W, \mu)^2 + \frac{\sigma^2}{n^2} M^2$$

In these inequalities we use the fact that $\min_W \max_\mu B(W, \mu)^2 = (\min_W \max_\mu B(W, \mu))^2$ and $\max_\mu \min_W B(W, \mu)^2 = (\max_\mu \min_W B(W, \mu))^2$ due to the symmetry of $\mu$ in $\mathcal{F}$ implied by Assumption 5.

## A.5 Proof of Lemma 4

Let $\prod$ denote Cartesian product. First we note that without loss of generality we can prove this lemma in the case that $\mathcal{F} = \prod_t \mathcal{F}_t$, since in general $F \subseteq \prod_t \mathcal{F}_t$ so $\sup_{\mu \in \mathcal{F}} \inf_W J(W, \mu) \leqslant \sup_{\mu \in \prod_t \mathcal{F}_t} \inf_W J(W, \mu)$, and it is easy to verify that all of our assumptions would still hold on the larger set $\prod_t \mathcal{F}_t$.

Now define the set $\mathcal{H}^0 = \{\mu \in \prod_t \mathcal{L}^2(\mathcal{Z}) : \mathbb{E}[\nu_T(X, T)^2] = 0\}$. Each coordinate $\mathcal{H}_t^0$ of $\mathcal{H}^0$ is a subspace of $\mathcal{L}^2(\mathcal{Z})$, so we can also define its orthogonal complement $\mathcal{H}_t^+$. Also, we have separability since from Section 2.1 we know that $\mathcal{Z} \subseteq \mathbb{R}^q$, so any function $f \in \mathcal{L}^2(\mathcal{Z})$ can be uniquely represented as $f = f^0 + f^+$ where $f^0 \in \mathcal{H}_t^0$ and $f^+ \in \mathcal{H}_t^+$. This means that for each $\mu_t \in \mathcal{F}_t$, we can similarly uniquely represent $\mu_t = \mu_t^0 + \mu_t^+$, and we can easily extend this to a unique representation of the vector $\mu = \mu^0 + \mu^+$. Now in the case that $\mathbb{E}[\nu_T(X, T)^2] = 0$ we have $\nu_{T_i}(X_i, T_i) = 0$ almost surely for all $i$, and it follows from Lemma 2 that $\nu_{T_i}(X_i, t) = 0$ almost surely also for all $i$ and $t$. Therefore any component of $\mu$ in $\mathcal{H}^0$ has no effect on the function $J(W, \mu)$ which we are bounding, so without loss of generality we can restrict our attention to the following space:

$$\mathcal{F}^+ = \prod_t (\mathcal{H}_t^+ \cap \mathcal{F}_t).$$

By construction the only function in $\mathcal{F}^+$ such that $\mathbb{E}[\nu_T(X, T)^2] = 0\}$ is the zero function, which we can also ignore in our bounds below, since when $\mu = 0$ we can easily obtain $J(W, \mu) = 0$ by choosing $W = 0$. Furthermore, by Assumption 7 we know that for each $t$ the $\mathcal{F}_t$ norm dominates the $\mathcal{L}^2(\mathcal{Z})$ norm, so it must be the case that that each space $\mathcal{F}_t^+$ is closed, since $\mathcal{H}_t^+$ is a closed subspace of $\mathcal{L}^2(\mathcal{Z})$ due to it being an orthogonal complement. Thus it follows easily from Assumption 4 that $\mathcal{F}^+$ is closed, given that its norm is an $\mathbb{R}^m$ norm on top of the corresponding $\mathcal{F}_t^+$ norms and $m$ is finite.

Now, based on Lemma 3, it is sufficient to pick weights in response to $\mu$ that control for a single mean outcome function. Instead of actually constructing a particular set of weights, we take the approach of viewing this as a convex optimization problem. Specifically, given $\mu$, we calculate the minimum euclidean norm of all weights that set the bias term $B(W, \mu)$ to zero exactly. This can be formulated as the following convex optimization program

$$\min_W \sum_i W_i^2$$

$$\text{s.t.} \sum_i W_i \nu_{T_i}(X_i, T_i) = \sum_{i,t} \pi_t(X_i) \nu_t(X_i, T_i).$$

Given the program only has linear constraints with equality, it satisfies Slater's condition, and therefore satisfies strong duality, which we will use to find the optimal value of this program. First we calculate the Lagrangian as

$$\mathcal{L}_n(W, \lambda) = \sum_i W_i^2 + \lambda \left( \sum_i W_i \nu_{T_i}(X_i, T_i) - \sum_{i,t} \pi_t(X_i) \nu_t(X_i, T_i) \right).$$

It can easily be verified by taking derivatives that for any $\lambda \in \mathbb{R}$ this function is minimized by setting $W_i = -\frac{\lambda}{2} \nu_{T_i}(X_i, T_i)$, and plugging this value in gives the dual formulation of the program as

$$\max_\lambda -\frac{\lambda^2}{4} \sum_i \nu_{T_i}(X_i, T_i)^2 - \lambda \sum_{i,t} \pi_t(X_i) \nu_t(X_i, T_i),$$

which is unconstrained. Again by taking derivatives we can maximize this function, and we find the maximum value is given by

$$\lambda = -2 \frac{\sum_i \pi_t(X_i) \nu_t(X_i, T_i)}{\sum_i \nu_{T_i}(X_i, T_i)^2},$$

and finally plugging this value into the dual objective function we see that the euclidean norm of the weights $W^*$ solving the convex program above is given by

$$\|W^*\|_2^2 = \frac{\left( \sum_i \pi_t(X_i) \nu_t(X_i, T_i) \right)^2}{\sum_i \nu_{T_i}(X_i, T_i)^2}.$$

Now define $\mathbb{E}_n$ as the mean with respect to the empirical distribution of the logged data. Then this objective value can be reformulated as

$$\|W^*\|_2^2 = \frac{n \mathbb{E}_n[\sum_t \nu_t(X, T)]^2}{\mathbb{E}_n[\nu_T(X, T)^2]}.$$

Therefore choosing $M = \|W^*\|$, combining this result with Lemma 3 gives us

$$\min_W \sup_{\mu \in \mathcal{F}^+} J(W, \mu) \leqslant \sup_{\mu \in \mathcal{F}^+} \frac{1}{n} \left( \frac{\sigma^2 \mathbb{E}_n[\sum_t \nu_t(X, T)]^2}{\mathbb{E}_n[\nu_T(X, T)^2]} \right)$$

$$= \sup_{\mu \in \mathcal{F}^+} \frac{1}{n} \left( \frac{\sigma^2 \mathbb{E}_n[\sum_t \nu_t(X, T)]^2}{\mathbb{E}[\nu_T(X, T)^2] + (\mathbb{E}_n[\nu_T(X, T)^2] - \mathbb{E}[\nu_T(X, T)^2])} \right).$$

Given this we will proceed by arguing that we can bound the denominator away from zero. We can note that $\mu$ appears in both the numerator and denominator on the same scale, so without loss of generality we can further restrict our attention to $\mu$ with fixed norm. By Assumption 8 we know that we can rescale every $\mu \in \mathcal{F}$ to have norm $\gamma$ for some $\gamma > 0$. Given this we will restrict ourselves to the set $\mathcal{F}_\gamma^+ = \{\mu \in \mathcal{F}^+ : \|\mu\| = \gamma\}$. Since $\mathcal{F}_\gamma^+$ is the intersection of two closed sets it must be closed. Furthermore by Assumption 6 it is also compact, so it satisfies the conditions for the extreme value theorem. By construction $\mathbb{E}[\nu_T(X, T)^2] > 0$ for every $\mu \in \mathcal{F}_\gamma^+$, so putting the above together we have $\inf_{\mu \in \mathcal{F}_\gamma^+} \mathbb{E}[\nu_T(X, T)^2] > 0$. We will define this value to be $\alpha$.

Now the numerator in the above bound is clearly bounded above by some $\beta > 0$ uniformly over $\mu \in \mathcal{F}$, since by Assumption 9 we know that every $\mu_t \in \mathcal{F}_t$ is uniformly bounded by some global constant, and therefore all $\nu$ terms are bounded by some constant $b$. Given this all that remains to be shown is that $\sup_{\mu \in \mathcal{F}} |\mathbb{E}_n[\nu_T(X, T)^2] - \mathbb{E}[\nu_T(X, T)^2]|$ converges in probability to zero. In order to show this we will define the following terms:

$$D_n = \mathbb{E}_n[\nu_T(X, T)^2]$$
$$E_n = \sup_{\mu \in \mathcal{F}} |D_n - \mathbb{E}[D_n]|$$

We need to show that $E_n$ converges uniformly to zero. Define $D_n'$ as an arbitrary recalculation of $D_n$ replacing $(X_{1:n}, T_{1:n})$ with $(X_{1:n}', T_{1:n}')$, which differ from the originals at most in a single

coordinate $i$, and define $E'_n = \sup_{\mu \in \mathcal{F}} |D'_n - \mathbb{E}[D'_n]|$. Furthermore as argued above all $\nu$ terms are bounded above by some constant $b$, so each $\nu_{T_i}(X_i, T_i)^2$ is bounded by $b^2$. Given this we can obtain

$$
\begin{aligned}
|E_n - E'_n| &= |\sup_{\mu \in \mathcal{F}} |D_n - \mathbb{E}[D_n]| - \sup_{\mu \in \mathcal{F}} |D'_n - \mathbb{E}[D'_n]||\\
&\leqslant \sup_{\mu \in \mathcal{F}} |(D_n - \mathbb{E}[D_n]) - (D'_n - \mathbb{E}[D'_n])|\\
&= \sup_{\mu \in \mathcal{F}} |D_n - D'_n|\\
&= \frac{1}{n} \sup_{\mu \in \mathcal{F}} |\nu_{T_i}(X_i, T_i)^2 - \nu_{T'_i}(X'_i, T'_i)^2|\\
&\leqslant \frac{2b^2}{n}
\end{aligned}
$$

Given this we can apply McDiarmid's inequality to obtain the following bound:

$$
P(|E_n - \mathbb{E}[E_n]| \leqslant \epsilon) \leqslant 2 \exp\left(-\frac{n\epsilon^2}{2b^4}\right)
$$

This implies that $E_n - \mathbb{E}[E_n] = o_p(1)$. Next we show that $\mathbb{E}[E_n] = o_p(1)$ also. We do this using a symmetrization argument as follows, where $D'_n$ is defined identically to $D_n$ using iid shadow variables $(X'_i, T'_i)$ in place of $(X_i, T_i)$ for each $i$, and $\epsilon_i$ are iid Rademacher random variables:

$$
\begin{aligned}
\mathbb{E}[E_n] &= \mathbb{E}\left[\sup_{\mu \in \mathcal{F}} |D_n - \mathbb{E}[D_n]|\right]\\
&= \mathbb{E}\left[\sup_{\mu \in \mathcal{F}} \left|\frac{1}{n}\sum_i \nu_{T_i}(X_i, T_i) - \mathbb{E}[\nu_{T'_i}(X'_i, T'_i)]\right|\right]\\
&\leqslant 2\mathbb{E}\left[\sup_{\mu \in \mathcal{F}} \left|\frac{1}{n}\sum_i \epsilon_i \nu_{T_i}(X_i, T_i)\right|\right]\\
&\leqslant 2\sum_t \mathbb{E}\left[\sup_{\mu \in \mathcal{F}} \left|\frac{1}{n}\sum_i \epsilon_i \delta_{T_i t} \nu_t(X_i, T_i)\right|\right]\\
&\leqslant 4\sum_t \mathbb{E}\left[\sup_{\mu \in \mathcal{F}} \left|\frac{1}{n}\sum_i \epsilon_i \nu_t(X_i, T_i)\right|\right]\\
&\leqslant 4\sum_t \mathbb{E}\left[\sup_{\mu \in \mathcal{F}} \left|\frac{1}{n}\sum_i \epsilon_i \mu_t(Z_i)\right|\right]\\
&\leqslant 4\sum_t \mathcal{R}_n(\mathcal{F}_t)
\end{aligned}
$$

where in the third inequality we appeal to the Rademacher comparison lemma [28, Thm. 4.12]. Thus since from Assumption 10 we know that the Rademacher complexity of each set $\mathcal{R}_n(\mathcal{F}_n)$ vanishes, it follows that $\mathbb{E}[E_n] = o_p(1)$. Putting everything from above together we get

$$
\min_W \sup_{\mu \in \mathcal{F}} J(W, \mu) \leqslant \frac{1}{n}\frac{\beta}{\alpha + o_p(1)},
$$

so we have $\min_W \sup_{\mu \in \mathcal{F}} J(W, \mu) \leqslant O_p(1/n)$ as required.

### A.6 Proof of Theorem 4

First, Assumption 4 follows trivially from the definition of $\mathcal{F}^K$. Next, Assumption 5 and *Assumption* 8 follow from the fact that $\mathcal{F}^K$ consists of all functions in $\text{span}(\mathcal{F}_t)$ with norm at most 1, as does the fact that it is a closed space. Given that $K$ is a Mercer kernel, balls in the corresponding RKHS have finite covering number [44], and it follows easily from this that $\mathcal{F}^K$ has

finite covering numbers, as its covering number must be bounded above by the sum of the covering numbers of the spaces $\mathcal{F}_t^K$. So $\mathcal{F}^K$ is closed and totally bounded with respect to its norm, and therefore compact, which gives us Assumption 6. Clearly each $\mathcal{F}_t$ is contained in $\mathcal{L}^2(\mathcal{Z})$ since RKHSs are square integrable, and the fact that the $K$ norm dominates the $\mathcal{L}^2$ follows from Mercer's Theorem, which implies that $\|f\|_K^2 = \sum_{i=1}^{\infty} f_i^2/\sigma_i$, where $f_i$ is the $i$th eigenvalue of $f$ for some orthonormal basis of $\mathcal{L}^2(\mathcal{Z})$, and $\sigma_i \geqslant 0$ converges to zero. This gives us Assumption 7. Next, by construction each $\mathcal{F}_t^K$ consists of all functions in the RKHS up to norm 1. Therefore assuming the kernel $K$ is bounded, it is trivial to verify that function application must be globally bounded, since for any function $f \in \mathcal{F}_t^K$ we have $f(x) \leqslant <f, K_x> \leqslant \|f\|\sqrt{K(x,x)} \leqslant \sqrt{K(x,x)}$, which gives us Assumption 9. Finally, given this characterization of $\mathcal{F}_t^K$ as the 1-ball of the RKHS, it has vanishing Rademacher complexity [33, Thm. 2.1], so we have Assumption 10.

### A.7 Proof of Theorem 5

First we will find a closed form expression for $\sup_{\mu \in \mathcal{F}^K} (\frac{1}{n} \sum_{i,t} f_{it}\nu_t(X_i, T_i))^2$, where $\nu_t$ is defined in terms of the approximate posterior $\hat{\varphi}$. In this derivation we will use the shorthand $\varphi_i$ for the conditional density of $Z_i$ given $X_i$ and $T_i$ under $\hat{\varphi}$, and $T_K$ for the kernel intergral operator defined according to $T_K f = \int_{\mathcal{Z}} K(\cdot, z)f(z)dz$. In this derivation we will make use of the fact that $\langle f, g \rangle_{\mathcal{L}^2} = \langle f, T_K g \rangle_K$ for any square integrable $f$ and $g$. Given all this we can obtain:

$$
\begin{aligned}
\sup_{\mu \in \mathcal{F}^K} \left( \frac{1}{n} \sum_{i,t} f_{it}\nu_t(X_i, T_i) \right)^2 &= \sum_t \sup_{\mu_t \in \mathcal{F}_t^K} \left( \frac{1}{n} \sum_i f_{it} \langle \mu_t, \varphi_i \rangle_{\mathcal{L}^2} \right)^2 \\
&= \sum_t \sup_{\mu_t \in \mathcal{F}_t^K} \left\langle \mu_t, \frac{1}{n} \sum_i f_{it}\varphi_i \right\rangle_{\mathcal{L}^2}^2 \\
&= \sum_t \sup_{\mu_t \in \mathcal{F}_t^K} \left\langle \mu_t, T_K \frac{1}{n} \sum_i f_{it}\varphi_i \right\rangle_K^2 \\
&= \sum_t \frac{\left\langle T_K \frac{1}{n} \sum_i f_{it}\varphi_i, T_K \frac{1}{n} \sum_i f_{it}\varphi_i \right\rangle_K^2}{\|T_K \frac{1}{n} \sum_i f_{it}\varphi_i\|_K} \\
&= \sum_t \left\langle T_K \frac{1}{n} \sum_i f_{it}\varphi_i, T_K \frac{1}{n} \sum_i f_{it}\varphi_i \right\rangle_K \\
&= \sum_t \left\langle \frac{1}{n} \sum_i f_{it}\varphi_i, T_K \frac{1}{n} \sum_i f_{it}\varphi_i \right\rangle_{\mathcal{L}^2} \\
&= \frac{1}{n^2} \sum_{i,j,t} f_{it}f_{jt} \langle \varphi_i, T_K \varphi_j \rangle_{\mathcal{L}^2} \\
&= \frac{1}{n^2} \sum_{i,j,t} f_{it}f_{jt} \int_{\mathcal{Z}} \varphi_i(z)(\int_{\mathcal{Z}'} K(z,z')\varphi_j(z')dz')dz \\
&= \frac{1}{n^2} \sum_{i,j,t} f_{it}f_{jt} \int_{\mathcal{Z}} \int_{\mathcal{Z}'} \varphi_i(z)\varphi_j(z')K(z,z')dz'dz \\
&= \frac{1}{n^2} \sum_{i,j,t} f_{it}f_{jt}\mathbb{E}[K(Z_i, Z_j')],
\end{aligned}
$$

where the expectation in the last term is implicitly defined in terms of the approximate posterior $\hat{\varphi}$ and conditional on the observed data.

Given this, and recalling that $f_{it} = W_i \delta_{T_i t} - \pi_t(X_i)$ we can derive a closed form for our minimization objective, as follows:

$$\sup_{\mu \in \mathcal{F}^K} J(W, \mu) = \frac{1}{n^2} \sum_{i,j,t} f_{it} f_{jt} Q_{ij} + \frac{1}{n^2} \sum_{i,j} W_i W_j \Gamma_{ij}$$

$$= \frac{1}{n^2} \sum_{i,j,t} Q_{ij} (W_i W_j \delta_{T_i t} \delta_{T_j t} - 2 W_j \delta_{T_j t} \pi_t(X_i) + \pi_t(X_i) \pi_t(X_j))$$

$$+ \frac{1}{n^2} \sum_{i,j} W_i W_j \Gamma_{ij}$$

$$= \frac{1}{n^2} \sum_{i,j} W_i W_j (Q_{ij} \delta_{T_i T_j} + \Gamma_{ij}) - \frac{2}{n^2} \sum_j W_j (\sum_i Q_{ij} \pi_{T_j}(X_i))$$

$$+ \frac{1}{n^2} \sum_{i,j,t} Q_{ij} \pi_t(X_i) \pi_t(X_j)$$

Finally we can conclude by noting that this corresponds to the quadratic program formulation given in the question with $c = \frac{1}{n^2} \sum_{i,j,t} Q_{ij} \pi_t(X_i) \pi_t(X_j)$.

# B   Additional Experimentation Details

## B.1   Experiment Scenario

All our experiments were conducted using the setup described in Section 5.1. We used the following parameter values for our data-generating distribution:

$$\alpha = [1.0, -2.0, -1.0, 2.0, 4.0, 0.0, -2.0, -1.0, -3.0, 1.0]$$
$$\alpha_0 = 0$$
$$\sigma_X = 4.0$$
$$\beta = [0.5, -0.5]$$
$$\beta_0 = 0$$
$$\zeta(0) = 1.0$$
$$\zeta(0)_0 = 0$$
$$\zeta(1) = -0.5$$
$$\zeta(1)_0 = 0$$
$$\sigma_Y = 0.01$$

In addiiton, the policy $\pi$ we are evaluating takes the form as described in Section 5.1, and we used the following parameter values for this policy:

$$\psi_0 = [-0.1, 0.2, 0.2, -0.1, -0.1, -0.1, 0.1, 0.1, 0.1, -0.1]$$
$$\psi_1 = -\psi_0$$

## B.2   Method Implementation Details

In all methods where we sampled from the posterior $\hat{\wp}(\cdot; x, t)$, this sampling was done using STAN [7], solving QPs and LCQPs was done using the Python package quadprog,[3] all stochastic gradient descent (SGD) learning was performed using the Adam [25] optimizer with a learning rate of 0.001.

**OptZ**   We ran Algorithm 1 with $B = 50$.

**IPS**   Since the propensity scores $\eta_t(x)$ are not not tractable to compute analytically, we trained a neural network $\hat{\eta}$ to estimate this function. This was done by sampling batches of $(Z, X)$ pairs

from the data model, and training the network using SGD to predict the vector of probabilities $P_T = \beta^T Z + \beta_0$ from $X$, using cross-entropy loss. We used a neural network with two hidden layers of size 200 for $\hat{\eta}$, and trained for 2000 iterations with a batch size of 32. We found in practice this training was very stable and gave accurate results.

**DirX** For each $t$ we trained a neural network $\hat{\rho}_t$ to predict $\nu_t(x, t)$ by taking the set of $(X, T, Y)$ triplets in our training data where $T = t$, and training the network using SGD to predict $Y$ from $X$ using MSE loss. Based on pilot experiments we used a network architecture with a single hidden layer of size 100, and trained using a batch size of 128. We used $80\%$ of our data for training, and used the remaining $20\%$ for the purpose of early stopping. We trained for a maximum of 500 epochs, or until we made no progress on development data for 20 epochs.

**DirZ** For each $t$ we trained a neural network $\hat{\mu}_t$ to predict $\mu_t$. This was done by taking the set of $(X, T, Y)$ triplets in our training data, and for each sampling 200 $Z$ values from the posterior using our identified model given $X$ and $T$. This gives us a set of $(Z, T, Y)$ triplets 200 times as large as our original training set. We then trained each $\hat{\mu}_t$ network by taking the set of these triplets where $T = t$, and optimized the network using SGD on this data predicting $Y$ from $Z$. We used the same settings for this optimization as with the *direct-naive* method, except we allowed up to 1000 epochs. Note that for both training and inference we limited ourselves to sampling 200 $Z$ values per data point due to computational limitations.

# C   Additional Experiment Results

## C.1   Results with Alternative Link Functions and Constraints

We present here our additional experiment results. In these results **SimplexOptZ** refers to our method using the simplex constraints discussed in Section 4.2.

| n | $\mathbf{OptZ}_{0.001}$ | $\mathbf{OptZ}_{0.2}$ | $\mathbf{OptZ}_{1.0}$ | $\mathbf{OptZ}_{5.0}$ |
|---|---|---|---|---|
| 200 | $.39 \pm .07$ | $.24 \pm .02$ | $.36 \pm .02$ | $.81 \pm .02$ |
| 500 | $.19 \pm .02$ | $.18 \pm .02$ | $.23 \pm .02$ | $.49 \pm .02$ |
| 1000 | $.11 \pm .01$ | $.11 \pm .01$ | $.13 \pm .01$ | $.27 \pm .01$ |
| 2000 | $.08 \pm .01$ | $.08 \pm .01$ | $.09 \pm .01$ | $.17 \pm .01$ |

Table 5: Convergence of RMSE for weighted estimator using our weights, with **step** link

| n | $\mathbf{DirX:OptZ}_{0.001}$ | $\mathbf{DirX:OptZ}_{0.2}$ | $\mathbf{DirX:OptZ}_{1.0}$ | $\mathbf{DirX:OptZ}_{5.0}$ |
|---|---|---|---|---|
| 200 | $.57 \pm .06$ | $.42 \pm .03$ | $.39 \pm .03$ | $.43 \pm .03$ |
| 500 | $.55 \pm .02$ | $.46 \pm .02$ | $.39 \pm .02$ | $.37 \pm .02$ |
| 1000 | $.49 \pm .02$ | $.45 \pm .01$ | $.39 \pm .01$ | $.32 \pm .01$ |
| 2000 | $.48 \pm .01$ | $.47 \pm .01$ | $.42 \pm .01$ | $.34 \pm .01$ |

Table 6: Convergence of RMSE for doubly robust estimator using our weights and **DirX**, with **step** link

| n | $\mathbf{DirZ:OptZ}_{0.001}$ | $\mathbf{DirZ:OptZ}_{0.2}$ | $\mathbf{DirZ:OptZ}_{1.0}$ | $\mathbf{DirZ:OptZ}_{5.0}$ |
|---|---|---|---|---|
| 200 | $.41 \pm .07$ | $.29 \pm .02$ | $.50 \pm .02$ | $1.1 \pm .03$ |
| 500 | $.20 \pm .02$ | $.21 \pm .02$ | $.31 \pm .02$ | $.70 \pm .02$ |
| 1000 | $.11 \pm .01$ | $.13 \pm .01$ | $.18 \pm .01$ | $.42 \pm .01$ |
| 2000 | $.08 \pm .01$ | $.09 \pm .01$ | $.13 \pm .01$ | $.26 \pm .01$ |

Table 7: Convergence of RMSE for doubly robust estimator using our weights and **DirZ**, with **step** link

| n | $\mathbf{SimplexOptZ}_{0.001}$ | $\mathbf{SimplexOptZ}_{0.2}$ | $\mathbf{SimplexOptZ}_{1.0}$ | $\mathbf{SimplexOptZ}_{5.0}$ |
|---|---|---|---|---|
| 200 | $.30 \pm .02$ | $.25 \pm .02$ | $.38 \pm .02$ | $.91 \pm .02$ |
| 500 | $.18 \pm .02$ | $.19 \pm .02$ | $.24 \pm .02$ | $.54 \pm .02$ |
| 1000 | $.12 \pm .01$ | $.11 \pm .01$ | $.13 \pm .01$ | $.29 \pm .01$ |
| 2000 | $.07 \pm .01$ | $.08 \pm .01$ | $.10 \pm .01$ | $.18 \pm .01$ |

Table 8: Convergence of RMSE for weighted estimator using our weights and constraining $W \in n\Delta^n$, with **step** link

| n | $\mathbf{IPS}$ | $\mathbf{OptX}_{0.001}$ | $\mathbf{OptX}_{0.2}$ | $\mathbf{OptX}_{1.0}$ | $\mathbf{OptX}_{5.0}$ | $\mathbf{DirX}$ | $\mathbf{DirZ}$ |
|---|---|---|---|---|---|---|---|
| 200 | $.47 \pm .03$ | $2.0 \pm .03$ | $2.1 \pm .03$ | $2.3 \pm .02$ | $2.5 \pm .02$ | $.52 \pm .02$ | $2.6 \pm .02$ |
| 500 | $.48 \pm .03$ | $2.0 \pm .02$ | $2.1 \pm .02$ | $2.3 \pm .02$ | $2.6 \pm .02$ | $.48 \pm .02$ | $2.6 \pm .01$ |
| 1000 | $.39 \pm .02$ | $2.0 \pm .01$ | $2.1 \pm .01$ | $2.3 \pm .01$ | $2.5 \pm .01$ | $.48 \pm .02$ | $2.6 \pm .01$ |
| 2000 | $.40 \pm .01$ | $2.0 \pm .01$ | $2.1 \pm .01$ | $2.3 \pm .01$ | $2.5 \pm .01$ | $.45 \pm .02$ | $2.6 \pm .01$ |

Table 9: Convergence of RMSE for benchmark methods, with **step** link

| n | $\mathbf{OptZ}_{0.001}$ | $\mathbf{OptZ}_{0.2}$ | $\mathbf{OptZ}_{1.0}$ | $\mathbf{OptZ}_{5.0}$ |
|---|---|---|---|---|
| 200 | $.03 \pm .39$ | $.11 \pm .21$ | $.29 \pm .21$ | $.78 \pm .18$ |
| 500 | $.09 \pm .17$ | $.10 \pm .15$ | $.17 \pm .16$ | $.47 \pm .15$ |
| 1000 | $.02 \pm .11$ | $.05 \pm .09$ | $.08 \pm .09$ | $.25 \pm .09$ |
| 2000 | $.03 \pm .07$ | $.05 \pm .06$ | $.07 \pm .07$ | $.16 \pm .07$ |

Table 10: Convergence of bias for weighted estimator using our weights, with **step** link

| n | $\mathbf{DirX:OptZ}_{0.001}$ | $\mathbf{DirX:OptZ}_{0.2}$ | $\mathbf{DirX:OptZ}_{1.0}$ | $\mathbf{DirX:OptZ}_{5.0}$ |
|---|---|---|---|---|
| 200 | $.43 \pm .38$ | $.35 \pm .24$ | $.31 \pm .24$ | $.37 \pm .22$ |
| 500 | $.51 \pm .19$ | $.42 \pm .18$ | $.35 \pm .18$ | $.33 \pm .17$ |
| 1000 | $.47 \pm .13$ | $.44 \pm .11$ | $.37 \pm .10$ | $.30 \pm .11$ |
| 2000 | $.47 \pm .09$ | $.46 \pm .08$ | $.41 \pm .08$ | $.33 \pm .08$ |

Table 11: Convergence of bias for doubly robust estimator using our weights and **DirX**, with **step** link

| n | $\mathbf{DirZ:OptZ}_{0.001}$ | $\mathbf{DirZ:OptZ}_{0.2}$ | $\mathbf{DirZ:OptZ}_{1.0}$ | $\mathbf{DirZ:OptZ}_{5.0}$ |
|---|---|---|---|---|
| 200 | $.05 \pm .40$ | $.19 \pm .22$ | $.45 \pm .22$ | $1.1 \pm .21$ |
| 500 | $.10 \pm .18$ | $.14 \pm .16$ | $.26 \pm .16$ | $.68 \pm .17$ |
| 1000 | $.04 \pm .11$ | $.09 \pm .10$ | $.15 \pm .10$ | $.41 \pm .10$ |
| 2000 | $.03 \pm .07$ | $.06 \pm .07$ | $.10 \pm .07$ | $.25 \pm .07$ |

Table 12: Convergence of bias for doubly robust estimator using our weights and **DirZ**, with **step** link

| n | $\mathbf{SimplexOptZ}_{0.001}$ | $\mathbf{SimplexOptZ}_{0.2}$ | $\mathbf{SimplexOptZ}_{1.0}$ | $\mathbf{SimplexOptZ}_{5.0}$ |
|---|---|---|---|---|
| 200 | $.04 \pm .30$ | $.12 \pm .21$ | $.31 \pm .21$ | $.89 \pm .20$ |
| 500 | $.08 \pm .15$ | $.10 \pm .15$ | $.18 \pm .16$ | $.51 \pm .16$ |
| 1000 | $.01 \pm .12$ | $.06 \pm .09$ | $.09 \pm .09$ | $.27 \pm .10$ |
| 2000 | $.03 \pm .07$ | $.05 \pm .06$ | $.07 \pm .07$ | $.17 \pm .07$ |

Table 13: Convergence of bias for weighted estimator using our weights and constraining $W \in n\Delta^n$, with **step** link

| n | **IPS** | $\mathbf{OptX}_{0.001}$ | $\mathbf{OptX}_{0.2}$ | $\mathbf{OptX}_{1.0}$ | $\mathbf{OptX}_{5.0}$ | **DirX** | **DirZ** |
|---|---|---|---|---|---|---|---|
| 200 | $.40 \pm .25$ | $1.9 \pm .21$ | $2.1 \pm .20$ | $2.3 \pm .19$ | $2.5 \pm .18$ | $.49 \pm .18$ | $2.6 \pm .14$ |
| 500 | $.43 \pm .21$ | $2.0 \pm .16$ | $2.1 \pm .15$ | $2.3 \pm .14$ | $2.6 \pm .13$ | $.45 \pm .16$ | $2.6 \pm .12$ |
| 1000 | $.37 \pm .12$ | $2.0 \pm .10$ | $2.1 \pm .09$ | $2.3 \pm .09$ | $2.5 \pm .08$ | $.46 \pm .15$ | $2.6 \pm .11$ |
| 2000 | $.39 \pm .10$ | $2.0 \pm .08$ | $2.1 \pm .07$ | $2.3 \pm .07$ | $2.5 \pm .07$ | $.42 \pm .17$ | $2.6 \pm .11$ |

Table 14: Convergence of bias for benchmark methods, with **step** link

| n | $\text{OptZ}_{0.001}$ | $\text{OptZ}_{0.2}$ | $\text{OptZ}_{1.0}$ | $\text{OptZ}_{5.0}$ |
|---|---|---|---|---|
| 200 | $.07 \pm .01$ | $.04 \pm .00$ | $.04 \pm .00$ | $.07 \pm .00$ |
| 500 | $.04 \pm .00$ | $.03 \pm .00$ | $.03 \pm .00$ | $.04 \pm .00$ |
| 1000 | $.02 \pm .00$ | $.02 \pm .00$ | $.02 \pm .00$ | $.02 \pm .00$ |
| 2000 | $.01 \pm .00$ | $.01 \pm .00$ | $.01 \pm .00$ | $.01 \pm .00$ |

Table 15: Convergence of RMSE for weighted estimator using our weights, with **exp** link

| n | $\text{DirX:OptZ}_{0.001}$ | $\text{DirX:OptZ}_{0.2}$ | $\text{DirX:OptZ}_{1.0}$ | $\text{DirX:OptZ}_{5.0}$ |
|---|---|---|---|---|
| 200 | $.13 \pm .01$ | $.10 \pm .01$ | $.12 \pm .01$ | $.11 \pm .01$ |
| 500 | $.10 \pm .01$ | $.09 \pm .01$ | $.10 \pm .01$ | $.12 \pm .01$ |
| 1000 | $.08 \pm .00$ | $.08 \pm .00$ | $.08 \pm .00$ | $.10 \pm .00$ |
| 2000 | $.07 \pm .00$ | $.07 \pm .00$ | $.08 \pm .00$ | $.09 \pm .00$ |

Table 16: Convergence of RMSE for doubly robust estimator using our weights and **DirX**, with **exp** link

| n | $\text{DirZ:OptZ}_{0.001}$ | $\text{DirZ:OptZ}_{0.2}$ | $\text{DirZ:OptZ}_{1.0}$ | $\text{DirZ:OptZ}_{5.0}$ |
|---|---|---|---|---|
| 200 | $.15 \pm .02$ | $.15 \pm .01$ | $.25 \pm .01$ | $.44 \pm .01$ |
| 500 | $.10 \pm .01$ | $.11 \pm .01$ | $.18 \pm .01$ | $.32 \pm .01$ |
| 1000 | $.07 \pm .01$ | $.08 \pm .01$ | $.12 \pm .01$ | $.23 \pm .01$ |
| 2000 | $.04 \pm .00$ | $.05 \pm .00$ | $.08 \pm .00$ | $.16 \pm .00$ |

Table 17: Convergence of RMSE for doubly robust estimator using our weights and **DirZ**, with **exp** link

| n | $\text{SimplexOptZ}_{0.001}$ | $\text{SimplexOptZ}_{0.2}$ | $\text{SimplexOptZ}_{1.0}$ | $\text{SimplexOptZ}_{5.0}$ |
|---|---|---|---|---|
| 200 | $.05 \pm .00$ | $.11 \pm .01$ | $.22 \pm .01$ | $.39 \pm .02$ |
| 500 | $.04 \pm .00$ | $.08 \pm .01$ | $.15 \pm .01$ | $.28 \pm .01$ |
| 1000 | $.02 \pm .00$ | $.05 \pm .00$ | $.09 \pm .00$ | $.19 \pm .00$ |
| 2000 | $.02 \pm .00$ | $.03 \pm .00$ | $.07 \pm .00$ | $.14 \pm .00$ |

Table 18: Convergence of RMSE for weighted estimator using our weights and constraining $W \in n\Delta^n$, with **exp** link

| n | IPS | $\text{OptX}_{0.001}$ | $\text{OptX}_{0.2}$ | $\text{OptX}_{1.0}$ | $\text{OptX}_{5.0}$ | DirX | DirZ |
|---|---|---|---|---|---|---|---|
| 200 | $.12 \pm .01$ | $.76 \pm .02$ | $.81 \pm .02$ | $.92 \pm .02$ | $1.0 \pm .02$ | $.10 \pm .01$ | $1.0 \pm .01$ |
| 500 | $.11 \pm .00$ | $.76 \pm .01$ | $.82 \pm .01$ | $.92 \pm .01$ | $1.0 \pm .01$ | $.10 \pm .01$ | $1.0 \pm .01$ |
| 1000 | $.10 \pm .00$ | $.74 \pm .01$ | $.79 \pm .01$ | $.90 \pm .01$ | $1.0 \pm .01$ | $.09 \pm .01$ | $1.1 \pm .01$ |
| 2000 | $.10 \pm .00$ | $.73 \pm .00$ | $.78 \pm .00$ | $.88 \pm .00$ | $.99 \pm .01$ | $.10 \pm .01$ | $1.0 \pm .01$ |

Table 19: Convergence of RMSE for benchmark methods, with **exp** link

| n | $\textbf{OptZ}_{0.001}$ | $\textbf{OptZ}_{0.2}$ | $\textbf{OptZ}_{1.0}$ | $\textbf{OptZ}_{5.0}$ |
|---|---|---|---|---|
| 200 | $.01 \pm .06$ | $.00 \pm .04$ | $-0.00 \pm .04$ | $-0.05 \pm .04$ |
| 500 | $.01 \pm .04$ | $.01 \pm .03$ | $.00 \pm .03$ | $-0.03 \pm .03$ |
| 1000 | $.00 \pm .02$ | $.01 \pm .02$ | $-0.00 \pm .02$ | $-0.01 \pm .02$ |
| 2000 | $.01 \pm .01$ | $.01 \pm .01$ | $.00 \pm .01$ | $.00 \pm .01$ |

Table 20: Convergence of bias for weighted estimator using our weights, with **exp** link

| n | $\textbf{DirX:OptZ}_{0.001}$ | $\textbf{DirX:OptZ}_{0.2}$ | $\textbf{DirX:OptZ}_{1.0}$ | $\textbf{DirX:OptZ}_{5.0}$ |
|---|---|---|---|---|
| 200 | $.07 \pm .10$ | $.08 \pm .06$ | $.10 \pm .05$ | $.11 \pm .04$ |
| 500 | $.07 \pm .07$ | $.07 \pm .05$ | $.09 \pm .05$ | $.11 \pm .04$ |
| 1000 | $.06 \pm .05$ | $.07 \pm .03$ | $.07 \pm .03$ | $.09 \pm .02$ |
| 2000 | $.06 \pm .04$ | $.06 \pm .03$ | $.07 \pm .03$ | $.09 \pm .03$ |

Table 21: Convergence of bias for doubly robust estimator using our weights and **DirX**, with **exp** link

| n | $\textbf{DirZ:OptZ}_{0.001}$ | $\textbf{DirZ:OptZ}_{0.2}$ | $\textbf{DirZ:OptZ}_{1.0}$ | $\textbf{DirZ:OptZ}_{5.0}$ |
|---|---|---|---|---|
| 200 | $.05 \pm .14$ | $.12 \pm .10$ | $.23 \pm .09$ | $.43 \pm .09$ |
| 500 | $.03 \pm .10$ | $.09 \pm .07$ | $.16 \pm .07$ | $.32 \pm .07$ |
| 1000 | $.02 \pm .06$ | $.06 \pm .05$ | $.10 \pm .05$ | $.23 \pm .06$ |
| 2000 | $.01 \pm .04$ | $.03 \pm .03$ | $.07 \pm .03$ | $.16 \pm .04$ |

Table 22: Convergence of bias for doubly robust estimator using our weights and **DirZ**, with **exp** link

| n | $\textbf{SimplexOptZ}_{0.001}$ | $\textbf{SimplexOptZ}_{0.2}$ | $\textbf{SimplexOptZ}_{1.0}$ | $\textbf{SimplexOptZ}_{5.0}$ |
|---|---|---|---|---|
| 200 | $.02 \pm .05$ | $.10 \pm .06$ | $.20 \pm .08$ | $.38 \pm .11$ |
| 500 | $.02 \pm .04$ | $.07 \pm .04$ | $.14 \pm .05$ | $.27 \pm .06$ |
| 1000 | $.01 \pm .02$ | $.04 \pm .02$ | $.08 \pm .03$ | $.19 \pm .04$ |
| 2000 | $.01 \pm .01$ | $.03 \pm .02$ | $.06 \pm .02$ | $.14 \pm .02$ |

Table 23: Convergence of bias for weighted estimator using our weights and constraining $W \in n\Delta^n$, with **exp** link

| n | **IPS** | $\textbf{OptX}_{0.001}$ | $\textbf{OptX}_{0.2}$ | $\textbf{OptX}_{1.0}$ | $\textbf{OptX}_{5.0}$ | **DirX** | **DirZ** |
|---|---|---|---|---|---|---|---|
| 200 | $.10 \pm .06$ | $.74 \pm .13$ | $.80 \pm .13$ | $.91 \pm .15$ | $1.0 \pm .16$ | $.07 \pm .07$ | $1.0 \pm .08$ |
| 500 | $.11 \pm .03$ | $.76 \pm .07$ | $.81 \pm .08$ | $.92 \pm .09$ | $1.0 \pm .10$ | $.07 \pm .08$ | $1.0 \pm .09$ |
| 1000 | $.10 \pm .03$ | $.74 \pm .05$ | $.79 \pm .05$ | $.90 \pm .06$ | $1.0 \pm .06$ | $.06 \pm .07$ | $1.0 \pm .10$ |
| 2000 | $.10 \pm .02$ | $.73 \pm .03$ | $.78 \pm .03$ | $.88 \pm .04$ | $.99 \pm .04$ | $.07 \pm .07$ | $1.0 \pm .09$ |

Table 24: Convergence of bias for benchmark methods, with **exp** link

| n | $\text{OptZ}_{0.001}$ | $\text{OptZ}_{0.2}$ | $\text{OptZ}_{1.0}$ | $\text{OptZ}_{5.0}$ |
|---|---|---|---|---|
| 200 | $.47 \pm .04$ | $.35 \pm .02$ | $.39 \pm .02$ | $.48 \pm .02$ |
| 500 | $.36 \pm .05$ | $.27 \pm .02$ | $.30 \pm .02$ | $.40 \pm .02$ |
| 1000 | $.25 \pm .02$ | $.22 \pm .01$ | $.25 \pm .01$ | $.37 \pm .01$ |
| 2000 | $.14 \pm .01$ | $.14 \pm .01$ | $.17 \pm .01$ | $.27 \pm .01$ |

Table 25: Convergence of RMSE for weighted estimator using our weights, with **cubic** link

| n | $\text{DirX:OptZ}_{0.001}$ | $\text{DirX:OptZ}_{0.2}$ | $\text{DirX:OptZ}_{1.0}$ | $\text{DirX:OptZ}_{5.0}$ |
|---|---|---|---|---|
| 200 | $.58 \pm .07$ | $.41 \pm .03$ | $.42 \pm .02$ | $.38 \pm .02$ |
| 500 | $.37 \pm .04$ | $.33 \pm .02$ | $.35 \pm .02$ | $.37 \pm .02$ |
| 1000 | $.31 \pm .02$ | $.31 \pm .02$ | $.33 \pm .02$ | $.39 \pm .01$ |
| 2000 | $.21 \pm .02$ | $.23 \pm .02$ | $.26 \pm .01$ | $.32 \pm .01$ |

Table 26: Convergence of RMSE for doubly robust estimator using our weights and **DirX**, with **cubic** link

| n | $\text{DirZ:OptZ}_{0.001}$ | $\text{DirZ:OptZ}_{0.2}$ | $\text{DirZ:OptZ}_{1.0}$ | $\text{DirZ:OptZ}_{5.0}$ |
|---|---|---|---|---|
| 200 | $.49 \pm .04$ | $.42 \pm .03$ | $.54 \pm .03$ | $.76 \pm .02$ |
| 500 | $.38 \pm .05$ | $.30 \pm .02$ | $.38 \pm .02$ | $.59 \pm .02$ |
| 1000 | $.27 \pm .02$ | $.25 \pm .02$ | $.32 \pm .02$ | $.52 \pm .02$ |
| 2000 | $.16 \pm .01$ | $.16 \pm .01$ | $.22 \pm .01$ | $.39 \pm .01$ |

Table 27: Convergence of RMSE for doubly robust estimator using our weights and **DirZ**, with **cubic** link

| n | $\text{SimplexOptZ}_{0.001}$ | $\text{SimplexOptZ}_{0.2}$ | $\text{SimplexOptZ}_{1.0}$ | $\text{SimplexOptZ}_{5.0}$ |
|---|---|---|---|---|
| 200 | $.45 \pm .04$ | $.41 \pm .03$ | $.52 \pm .03$ | $.70 \pm .03$ |
| 500 | $.37 \pm .05$ | $.30 \pm .02$ | $.37 \pm .02$ | $.55 \pm .02$ |
| 1000 | $.26 \pm .02$ | $.24 \pm .02$ | $.31 \pm .02$ | $.50 \pm .02$ |
| 2000 | $.14 \pm .01$ | $.15 \pm .01$ | $.20 \pm .01$ | $.35 \pm .01$ |

Table 28: Convergence of RMSE for weighted estimator using our weights and constraining $W \in n\Delta^n$, with **cubic** link

| n | IPS | $\text{OptX}_{0.001}$ | $\text{OptX}_{0.2}$ | $\text{OptX}_{1.0}$ | $\text{OptX}_{5.0}$ | DirX | DirZ |
|---|---|---|---|---|---|---|---|
| 200 | $.46 \pm .04$ | $1.1 \pm .02$ | $1.1 \pm .03$ | $1.3 \pm .03$ | $1.4 \pm .03$ | $.36 \pm .02$ | $1.4 \pm .01$ |
| 500 | $.38 \pm .02$ | $1.1 \pm .01$ | $1.2 \pm .01$ | $1.3 \pm .01$ | $1.4 \pm .01$ | $.34 \pm .02$ | $1.4 \pm .01$ |
| 1000 | $.39 \pm .01$ | $1.1 \pm .01$ | $1.2 \pm .01$ | $1.3 \pm .01$ | $1.4 \pm .01$ | $.35 \pm .02$ | $1.4 \pm .01$ |
| 2000 | $.35 \pm .01$ | $1.1 \pm .01$ | $1.2 \pm .01$ | $1.3 \pm .01$ | $1.4 \pm .01$ | $.39 \pm .02$ | $1.4 \pm .01$ |

Table 29: Convergence of RMSE for benchmark methods, with **cubic** link

| n | $\mathbf{OptZ}_{0.001}$ | $\mathbf{OptZ}_{0.2}$ | $\mathbf{OptZ}_{1.0}$ | $\mathbf{OptZ}_{5.0}$ |
|---|---|---|---|---|
| 200 | $.01 \pm .47$ | $.16 \pm .31$ | $.29 \pm .27$ | $.45 \pm .16$ |
| 500 | $-0.01 \pm .36$ | $.12 \pm .24$ | $.22 \pm .21$ | $.37 \pm .15$ |
| 1000 | $.03 \pm .25$ | $.12 \pm .18$ | $.20 \pm .15$ | $.35 \pm .12$ |
| 2000 | $.02 \pm .14$ | $.07 \pm .12$ | $.13 \pm .11$ | $.26 \pm .08$ |

Table 30: Convergence of bias for weighted estimator using our weights, with **cubic** link

| n | $\mathbf{DirX{:}OptZ}_{0.001}$ | $\mathbf{DirX{:}OptZ}_{0.2}$ | $\mathbf{DirX{:}OptZ}_{1.0}$ | $\mathbf{DirX{:}OptZ}_{5.0}$ |
|---|---|---|---|---|
| 200 | $.13 \pm .57$ | $.26 \pm .32$ | $.33 \pm .27$ | $.34 \pm .17$ |
| 500 | $.12 \pm .35$ | $.22 \pm .25$ | $.28 \pm .21$ | $.34 \pm .15$ |
| 1000 | $.18 \pm .26$ | $.24 \pm .19$ | $.29 \pm .15$ | $.37 \pm .11$ |
| 2000 | $.14 \pm .16$ | $.18 \pm .14$ | $.22 \pm .12$ | $.31 \pm .09$ |

Table 31: Convergence of bias for doubly robust estimator using our weights and **DirX**, with **cubic** link

| n | $\mathbf{DirZ{:}OptZ}_{0.001}$ | $\mathbf{DirZ{:}OptZ}_{0.2}$ | $\mathbf{DirZ{:}OptZ}_{1.0}$ | $\mathbf{DirZ{:}OptZ}_{5.0}$ |
|---|---|---|---|---|
| 200 | $.04 \pm .49$ | $.25 \pm .34$ | $.45 \pm .29$ | $.74 \pm .20$ |
| 500 | $.03 \pm .38$ | $.17 \pm .25$ | $.32 \pm .21$ | $.57 \pm .16$ |
| 1000 | $.04 \pm .26$ | $.16 \pm .19$ | $.28 \pm .16$ | $.50 \pm .13$ |
| 2000 | $.03 \pm .16$ | $.10 \pm .13$ | $.19 \pm .12$ | $.37 \pm .10$ |

Table 32: Convergence of bias for doubly robust estimator using our weights and **DirZ**, with **cubic** link

| n | $\mathbf{SimplexOptZ}_{0.001}$ | $\mathbf{SimplexOptZ}_{0.2}$ | $\mathbf{SimplexOptZ}_{1.0}$ | $\mathbf{SimplexOptZ}_{5.0}$ |
|---|---|---|---|---|
| 200 | $.02 \pm .45$ | $.22 \pm .35$ | $.40 \pm .34$ | $.65 \pm .28$ |
| 500 | $.01 \pm .37$ | $.15 \pm .26$ | $.29 \pm .23$ | $.52 \pm .18$ |
| 1000 | $.04 \pm .25$ | $.15 \pm .19$ | $.27 \pm .17$ | $.47 \pm .14$ |
| 2000 | $.02 \pm .14$ | $.08 \pm .13$ | $.17 \pm .11$ | $.34 \pm .09$ |

Table 33: Convergence of bias for weighted estimator using our weights and constraining $W \in n\Delta^n$, with **cubic** link

| n | $\mathbf{IPS}$ | $\mathbf{OptX}_{0.001}$ | $\mathbf{OptX}_{0.2}$ | $\mathbf{OptX}_{1.0}$ | $\mathbf{OptX}_{5.0}$ | $\mathbf{DirX}$ | $\mathbf{DirZ}$ |
|---|---|---|---|---|---|---|---|
| 200 | $.26 \pm .38$ | $1.1 \pm .20$ | $1.1 \pm .20$ | $1.2 \pm .21$ | $1.4 \pm .22$ | $.32 \pm .16$ | $1.4 \pm .12$ |
| 500 | $.31 \pm .22$ | $1.1 \pm .10$ | $1.2 \pm .10$ | $1.3 \pm .10$ | $1.4 \pm .11$ | $.29 \pm .18$ | $1.4 \pm .10$ |
| 1000 | $.37 \pm .14$ | $1.1 \pm .07$ | $1.2 \pm .07$ | $1.3 \pm .08$ | $1.4 \pm .08$ | $.32 \pm .16$ | $1.4 \pm .10$ |
| 2000 | $.34 \pm .09$ | $1.1 \pm .05$ | $1.2 \pm .06$ | $1.3 \pm .06$ | $1.4 \pm .06$ | $.34 \pm .18$ | $1.4 \pm .11$ |

Table 34: Convergence of bias for benchmark methods, with **cubic** link

| n | $\mathbf{OptZ}_{0.001}$ | $\mathbf{OptZ}_{0.2}$ | $\mathbf{OptZ}_{1.0}$ | $\mathbf{OptZ}_{5.0}$ |
|---|---|---|---|---|
| 200 | $.09 \pm .01$ | $.08 \pm .01$ | $.13 \pm .01$ | $.23 \pm .01$ |
| 500 | $.06 \pm .01$ | $.06 \pm .00$ | $.09 \pm .00$ | $.16 \pm .00$ |
| 1000 | $.04 \pm .01$ | $.04 \pm .00$ | $.06 \pm .00$ | $.12 \pm .00$ |
| 2000 | $.02 \pm .00$ | $.03 \pm .00$ | $.04 \pm .00$ | $.08 \pm .00$ |

Table 35: Convergence of RMSE for weighted estimator using our weights, with **linear** link

| n | $\mathbf{DirX{:}OptZ}_{0.001}$ | $\mathbf{DirX{:}OptZ}_{0.2}$ | $\mathbf{DirX{:}OptZ}_{1.0}$ | $\mathbf{DirX{:}OptZ}_{5.0}$ |
|---|---|---|---|---|
| 200 | $.15 \pm .01$ | $.13 \pm .01$ | $.13 \pm .01$ | $.13 \pm .01$ |
| 500 | $.14 \pm .01$ | $.13 \pm .01$ | $.13 \pm .01$ | $.13 \pm .00$ |
| 1000 | $.13 \pm .01$ | $.13 \pm .00$ | $.13 \pm .00$ | $.13 \pm .00$ |
| 2000 | $.12 \pm .00$ | $.12 \pm .00$ | $.12 \pm .00$ | $.12 \pm .00$ |

Table 36: Convergence of RMSE for doubly robust estimator using our weights and **DirX**, with **linear** link

| n | $\mathbf{DirZ{:}OptZ}_{0.001}$ | $\mathbf{DirZ{:}OptZ}_{0.2}$ | $\mathbf{DirZ{:}OptZ}_{1.0}$ | $\mathbf{DirZ{:}OptZ}_{5.0}$ |
|---|---|---|---|---|
| 200 | $.11 \pm .01$ | $.11 \pm .01$ | $.18 \pm .01$ | $.35 \pm .01$ |
| 500 | $.06 \pm .01$ | $.08 \pm .01$ | $.13 \pm .01$ | $.24 \pm .01$ |
| 1000 | $.05 \pm .01$ | $.06 \pm .00$ | $.09 \pm .00$ | $.18 \pm .00$ |
| 2000 | $.03 \pm .00$ | $.04 \pm .00$ | $.06 \pm .00$ | $.12 \pm .00$ |

Table 37: Convergence of RMSE for doubly robust estimator using our weights and **DirZ**, with **linear** link

| n | $\mathbf{SimplexOptZ}_{0.001}$ | $\mathbf{SimplexOptZ}_{0.2}$ | $\mathbf{SimplexOptZ}_{1.0}$ | $\mathbf{SimplexOptZ}_{5.0}$ |
|---|---|---|---|---|
| 200 | $.09 \pm .01$ | $.09 \pm .01$ | $.15 \pm .01$ | $.29 \pm .01$ |
| 500 | $.06 \pm .01$ | $.07 \pm .01$ | $.10 \pm .01$ | $.19 \pm .01$ |
| 1000 | $.04 \pm .01$ | $.04 \pm .00$ | $.07 \pm .00$ | $.14 \pm .00$ |
| 2000 | $.02 \pm .00$ | $.03 \pm .00$ | $.05 \pm .00$ | $.09 \pm .00$ |

Table 38: Convergence of RMSE for weighted estimator using our weights and constraining $W \in n\Delta^n$, with **linear** link

| n | **IPS** | $\mathbf{OptX}_{0.001}$ | $\mathbf{OptX}_{0.2}$ | $\mathbf{OptX}_{1.0}$ | $\mathbf{OptX}_{5.0}$ | **DirX** | **DirZ** |
|---|---|---|---|---|---|---|---|
| 200 | $.15 \pm .01$ | $.57 \pm .01$ | $.60 \pm .01$ | $.66 \pm .01$ | $.72 \pm .01$ | $.13 \pm .00$ | $.76 \pm .00$ |
| 500 | $.15 \pm .01$ | $.57 \pm .00$ | $.60 \pm .00$ | $.66 \pm .00$ | $.72 \pm .00$ | $.13 \pm .00$ | $.76 \pm .00$ |
| 1000 | $.14 \pm .00$ | $.57 \pm .00$ | $.60 \pm .00$ | $.66 \pm .00$ | $.72 \pm .00$ | $.13 \pm .00$ | $.76 \pm .00$ |
| 2000 | $.14 \pm .00$ | $.57 \pm .00$ | $.60 \pm .00$ | $.66 \pm .00$ | $.72 \pm .00$ | $.13 \pm .00$ | $.76 \pm .00$ |

Table 39: Convergence of RMSE for benchmark methods, with **linear** link

| n | OptZ$_{0.001}$ | OptZ$_{0.2}$ | OptZ$_{1.0}$ | OptZ$_{5.0}$ |
|---|---|---|---|---|
| 200 | $.03 \pm .09$ | $.06 \pm .06$ | $.11 \pm .05$ | $.23 \pm .04$ |
| 500 | $.02 \pm .05$ | $.04 \pm .05$ | $.08 \pm .04$ | $.15 \pm .04$ |
| 1000 | $.01 \pm .04$ | $.03 \pm .03$ | $.05 \pm .03$ | $.11 \pm .03$ |
| 2000 | $.01 \pm .02$ | $.02 \pm .02$ | $.04 \pm .02$ | $.08 \pm .02$ |

Table 40: Convergence of bias for weighted estimator using our weights, with **linear** link

| n | DirX:OptZ$_{0.001}$ | DirX:OptZ$_{0.2}$ | DirX:OptZ$_{1.0}$ | DirX:OptZ$_{5.0}$ |
|---|---|---|---|---|
| 200 | $.12 \pm .09$ | $.12 \pm .06$ | $.12 \pm .05$ | $.13 \pm .04$ |
| 500 | $.13 \pm .06$ | $.12 \pm .05$ | $.12 \pm .04$ | $.12 \pm .04$ |
| 1000 | $.12 \pm .04$ | $.12 \pm .03$ | $.12 \pm .03$ | $.12 \pm .03$ |
| 2000 | $.12 \pm .03$ | $.12 \pm .03$ | $.12 \pm .02$ | $.12 \pm .02$ |

Table 41: Convergence of bias for doubly robust estimator using our weights and **DirX**, with **linear** link

| n | DirZ:OptZ$_{0.001}$ | DirZ:OptZ$_{0.2}$ | DirZ:OptZ$_{1.0}$ | DirZ:OptZ$_{5.0}$ |
|---|---|---|---|---|
| 200 | $.04 \pm .10$ | $.09 \pm .07$ | $.17 \pm .06$ | $.34 \pm .05$ |
| 500 | $.02 \pm .06$ | $.06 \pm .05$ | $.12 \pm .05$ | $.24 \pm .05$ |
| 1000 | $.02 \pm .04$ | $.05 \pm .04$ | $.08 \pm .04$ | $.18 \pm .04$ |
| 2000 | $.01 \pm .03$ | $.03 \pm .03$ | $.06 \pm .03$ | $.12 \pm .02$ |

Table 42: Convergence of bias for doubly robust estimator using our weights and **DirZ**, with **linear** link

| n | SimplexOptZ$_{0.001}$ | SimplexOptZ$_{0.2}$ | SimplexOptZ$_{1.0}$ | SimplexOptZ$_{5.0}$ |
|---|---|---|---|---|
| 200 | $.02 \pm .08$ | $.07 \pm .06$ | $.14 \pm .06$ | $.29 \pm .06$ |
| 500 | $.02 \pm .05$ | $.05 \pm .05$ | $.09 \pm .05$ | $.19 \pm .04$ |
| 1000 | $.01 \pm .04$ | $.03 \pm .03$ | $.06 \pm .03$ | $.14 \pm .03$ |
| 2000 | $.01 \pm .02$ | $.02 \pm .02$ | $.04 \pm .02$ | $.09 \pm .02$ |

Table 43: Convergence of bias for weighted estimator using our weights and constraining $W \in n\Delta^n$, with **linear** link

| n | IPS | OptX$_{0.001}$ | OptX$_{0.2}$ | OptX$_{1.0}$ | OptX$_{5.0}$ | DirX | DirZ |
|---|---|---|---|---|---|---|---|
| 200 | $.13 \pm .08$ | $.57 \pm .05$ | $.60 \pm .05$ | $.66 \pm .05$ | $.72 \pm .05$ | $.13 \pm .04$ | $.76 \pm .04$ |
| 500 | $.14 \pm .05$ | $.57 \pm .04$ | $.60 \pm .03$ | $.66 \pm .03$ | $.72 \pm .03$ | $.12 \pm .04$ | $.76 \pm .02$ |
| 1000 | $.14 \pm .04$ | $.57 \pm .02$ | $.60 \pm .02$ | $.66 \pm .02$ | $.72 \pm .02$ | $.13 \pm .03$ | $.76 \pm .03$ |
| 2000 | $.13 \pm .03$ | $.57 \pm .02$ | $.60 \pm .02$ | $.66 \pm .02$ | $.72 \pm .02$ | $.13 \pm .03$ | $.76 \pm .02$ |

Table 44: Convergence of bias for benchmark methods, with **linear** link

| n | $\mathbf{OptZ}_{0.001}$ | $\mathbf{OptZ}_{0.2}$ | $\mathbf{OptZ}_{1.0}$ | $\mathbf{OptZ}_{5.0}$ |
|---|---|---|---|---|
| 200 | $.94 \pm .21$ | $.31 \pm .04$ | $.52 \pm .04$ | $1.1 \pm .04$ |
| 500 | $.32 \pm .07$ | $.21 \pm .03$ | $.33 \pm .03$ | $.76 \pm .03$ |
| 1000 | $.23 \pm .04$ | $.13 \pm .02$ | $.18 \pm .01$ | $.48 \pm .01$ |
| 2000 | $.10 \pm .01$ | $.09 \pm .01$ | $.11 \pm .01$ | $.28 \pm .01$ |

Table 45: Convergence of RMSE for weighted estimator using our weights, with **step** link and $\dim_z = 2$

| n | IPS | $\mathbf{OptX}_{0.001}$ | $\mathbf{OptX}_{0.2}$ | $\mathbf{OptX}_{1.0}$ | $\mathbf{OptX}_{5.0}$ | DirX | DirZ |
|---|---|---|---|---|---|---|---|
| 200 | $.50 \pm .05$ | $2.2 \pm .04$ | $2.3 \pm .04$ | $2.5 \pm .04$ | $2.7 \pm .03$ | $.91 \pm .07$ | $2.8 \pm .04$ |
| 500 | $.50 \pm .03$ | $2.3 \pm .03$ | $2.4 \pm .03$ | $2.6 \pm .02$ | $2.8 \pm .02$ | $.57 \pm .04$ | $2.8 \pm .03$ |
| 1000 | $.47 \pm .02$ | $2.3 \pm .01$ | $2.4 \pm .01$ | $2.6 \pm .01$ | $2.8 \pm .01$ | $.50 \pm .04$ | $2.9 \pm .02$ |
| 2000 | $.42 \pm .02$ | $2.2 \pm .01$ | $2.3 \pm .01$ | $2.5 \pm .01$ | $2.8 \pm .01$ | $.40 \pm .02$ | $2.9 \pm .01$ |

Table 46: Convergence of RMSE for benchmark methods, with **step** link and $\dim_z = 2$

| n | $\mathbf{OptZ}_{0.001}$ | $\mathbf{OptZ}_{0.2}$ | $\mathbf{OptZ}_{1.0}$ | $\mathbf{OptZ}_{5.0}$ |
|---|---|---|---|---|
| 200 | $.30 \pm .89$ | $.17 \pm .25$ | $.46 \pm .23$ | $1.1 \pm .20$ |
| 500 | $.10 \pm .31$ | $.13 \pm .16$ | $.29 \pm .16$ | $.74 \pm .15$ |
| 1000 | $.02 \pm .22$ | $.07 \pm .11$ | $.15 \pm .10$ | $.47 \pm .09$ |
| 2000 | $-0.01 \pm .10$ | $.05 \pm .07$ | $.08 \pm .07$ | $.27 \pm .07$ |

Table 47: Convergence of bias for weighted estimator using our weights, with **step** link and $\dim_z = 2$

| n | IPS | $\mathbf{OptX}_{0.001}$ | $\mathbf{OptX}_{0.2}$ | $\mathbf{OptX}_{1.0}$ | $\mathbf{OptX}_{5.0}$ | DirX | DirZ |
|---|---|---|---|---|---|---|---|
| 200 | $.39 \pm .31$ | $2.2 \pm .25$ | $2.3 \pm .24$ | $2.5 \pm .22$ | $2.7 \pm .20$ | $.83 \pm .38$ | $2.8 \pm .26$ |
| 500 | $.46 \pm .20$ | $2.2 \pm .16$ | $2.3 \pm .15$ | $2.6 \pm .14$ | $2.8 \pm .13$ | $.53 \pm .21$ | $2.8 \pm .18$ |
| 1000 | $.45 \pm .11$ | $2.3 \pm .08$ | $2.4 \pm .07$ | $2.6 \pm .07$ | $2.8 \pm .06$ | $.45 \pm .21$ | $2.9 \pm .12$ |
| 2000 | $.40 \pm .11$ | $2.2 \pm .08$ | $2.3 \pm .08$ | $2.5 \pm .07$ | $2.7 \pm .07$ | $.38 \pm .11$ | $2.9 \pm .05$ |

Table 48: Convergence of bias for benchmark methods, with **step** link and $\dim_z = 2$

## C.2 Results with Varying Dimensionality of Hidden Confounders

We present some additional results here experimenting with varying the dimensionality of the hidden confounder $Z$. We experimented with increasing the dimensionality of $Z$ to either 2 or 5, while keeping the dimensionalities of all other components equal. We extended the data generating process components described in Appendix B.1 as follows:

- The $i'th$ row of $\alpha$ is defined by cyclically rotating the elements of the $\alpha$ defined in Appendix B.1 by $i - 1$ places clockwise.
- The $i'th$ row of $\beta$ is defined as $[0.5 - 0.05(i - 1), 0.5 + 0.05(i - 1)]$.
- The $i'th$ row of $\zeta(0)$ is defined as $1.0 - 0.3(i - 1)$.
- The $i'th$ row of $\zeta(1)$ is defined as $-0.5 + 0.3(i - 1)$.
- The extra rows of all other matrices that need to be extended to accommodate higher dimensional $Z$ are simply duplicated.

We present results below for our method and benchmarks, for the **step** link function. We can observe that the same overall pattern of behavior occurs, with our methods still appearing to be consistent, though with slower convergence as the dimensionality of $Z$ increases.

| n | OptZ$_{0.001}$ | OptZ$_{0.2}$ | OptZ$_{1.0}$ | OptZ$_{5.0}$ |
|---|---|---|---|---|
| 200 | .72 ± .07 | .42 ± .05 | .64 ± .04 | .92 ± .02 |
| 500 | .47 ± .07 | .31 ± .03 | .47 ± .03 | .79 ± .02 |
| 1000 | .35 ± .04 | .22 ± .02 | .36 ± .02 | .64 ± .01 |
| 2000 | .31 ± .05 | .17 ± .01 | .26 ± .01 | .47 ± .01 |

Table 49: Convergence of RMSE for weighted estimator using our weights, with **step** link and $\dim_z = 5$

| n | IPS | OptX$_{0.001}$ | OptX$_{0.2}$ | OptX$_{1.0}$ | OptX$_{5.0}$ | DirX | DirZ |
|---|---|---|---|---|---|---|---|
| 200 | .40 ± .06 | 1.1 ± .05 | 1.2 ± .04 | 1.3 ± .04 | 1.4 ± .04 | .44 ± .04 | 1.5 ± .05 |
| 500 | .33 ± .03 | 1.1 ± .03 | 1.2 ± .03 | 1.4 ± .03 | 1.5 ± .03 | .40 ± .04 | 1.5 ± .04 |
| 1000 | .29 ± .02 | 1.1 ± .02 | 1.2 ± .02 | 1.3 ± .02 | 1.5 ± .02 | .35 ± .03 | 1.6 ± .02 |
| 2000 | .26 ± .02 | 1.1 ± .01 | 1.1 ± .01 | 1.3 ± .01 | 1.4 ± .01 | .31 ± .02 | 1.5 ± .01 |

Table 50: Convergence of RMSE for benchmark methods, with **step** link and $\dim_z = 5$

| n | OptZ$_{0.001}$ | OptZ$_{0.2}$ | OptZ$_{1.0}$ | OptZ$_{5.0}$ |
|---|---|---|---|---|
| 200 | .03 ± .72 | .32 ± .27 | .61 ± .22 | .91 ± .13 |
| 500 | .01 ± .47 | .23 ± .21 | .44 ± .16 | .78 ± .11 |
| 1000 | −0.01 ± .35 | .18 ± .13 | .34 ± .11 | .64 ± .08 |
| 2000 | .03 ± .31 | .13 ± .11 | .24 ± .09 | .47 ± .07 |

Table 51: Convergence of bias for weighted estimator using our weights, with **step** link and $\dim_z = 5$

| n | IPS | OptX$_{0.001}$ | OptX$_{0.2}$ | OptX$_{1.0}$ | OptX$_{5.0}$ | DirX | DirZ |
|---|---|---|---|---|---|---|---|
| 200 | .23 ± .33 | 1.1 ± .25 | 1.1 ± .23 | 1.3 ± .21 | 1.4 ± .20 | .36 ± .25 | 1.5 ± .29 |
| 500 | .24 ± .23 | 1.1 ± .16 | 1.2 ± .16 | 1.4 ± .15 | 1.5 ± .15 | .32 ± .23 | 1.5 ± .21 |
| 1000 | .25 ± .15 | 1.1 ± .11 | 1.2 ± .11 | 1.3 ± .11 | 1.5 ± .10 | .31 ± .16 | 1.6 ± .10 |
| 2000 | .24 ± .12 | 1.1 ± .06 | 1.1 ± .06 | 1.3 ± .06 | 1.4 ± .06 | .29 ± .12 | 1.5 ± .07 |

Table 52: Convergence of bias for benchmark methods, with **step** link and $\dim_z = 5$

| n | $\mathbf{OptZ}_{0.001}$ | $\mathbf{OptZ}_{0.2}$ | $\mathbf{OptZ}_{1.0}$ | $\mathbf{OptZ}_{5.0}$ |
|---|---|---|---|---|
| 200 | .35 ± .05 | .20 ± .03 | .26 ± .02 | .70 ± .03 |
| 500 | .21 ± .02 | .12 ± .01 | .18 ± .01 | .38 ± .01 |
| 1000 | .13 ± .02 | .05 ± .01 | .11 ± .01 | .22 ± .01 |
| 2000 | .07 ± .01 | .03 ± .00 | .07 ± .01 | .16 ± .01 |

Table 53: Convergence of RMSE for weighted estimator using our weights, with **step** link and $\sigma_x = 0.1$

| n | IPS | $\mathbf{OptX}_{0.001}$ | $\mathbf{OptX}_{0.2}$ | $\mathbf{OptX}_{1.0}$ | $\mathbf{OptX}_{5.0}$ | DirX | DirZ |
|---|---|---|---|---|---|---|---|
| 200 | .18 ± .02 | .33 ± .03 | .44 ± .03 | .76 ± .03 | 1.6 ± .04 | .34 ± .06 | 3.2 ± .05 |
| 500 | .07 ± .01 | .18 ± .01 | .25 ± .01 | .45 ± .01 | 1.0 ± .02 | .08 ± .02 | 3.2 ± .04 |
| 1000 | .06 ± .01 | .09 ± .01 | .13 ± .01 | .26 ± .01 | .68 ± .01 | .04 ± .01 | 3.3 ± .02 |
| 2000 | .04 ± .00 | .05 ± .00 | .08 ± .00 | .16 ± .00 | .44 ± .01 | .04 ± .00 | 3.3 ± .01 |

Table 54: Convergence of RMSE for benchmark methods, with **step** link and $\sigma_x = 0.1$

| n | $\mathbf{OptZ}_{0.001}$ | $\mathbf{OptZ}_{0.2}$ | $\mathbf{OptZ}_{1.0}$ | $\mathbf{OptZ}_{5.0}$ |
|---|---|---|---|---|
| 200 | −0.12 ± .32 | .13 ± .15 | .23 ± .13 | .67 ± .17 |
| 500 | .11 ± .18 | .11 ± .06 | .17 ± .06 | .37 ± .06 |
| 1000 | .03 ± .13 | .02 ± .04 | .09 ± .05 | .21 ± .05 |
| 2000 | −0.03 ± .06 | .01 ± .03 | .06 ± .03 | .15 ± .03 |

Table 55: Convergence of bias for weighted estimator using our weights, with **step** link and $\sigma_x = 0.1$

| n | IPS | $\mathbf{OptX}_{0.001}$ | $\mathbf{OptX}_{0.2}$ | $\mathbf{OptX}_{1.0}$ | $\mathbf{OptX}_{5.0}$ | DirX | DirZ |
|---|---|---|---|---|---|---|---|
| 200 | −0.01 ± .18 | .29 ± .16 | .41 ± .16 | .74 ± .18 | 1.6 ± .22 | .17 ± .30 | 3.2 ± .29 |
| 500 | .01 ± .07 | .17 ± .07 | .24 ± .07 | .45 ± .07 | 1.0 ± .09 | .03 ± .07 | 3.2 ± .22 |
| 1000 | −0.02 ± .06 | .08 ± .04 | .12 ± .04 | .26 ± .05 | .67 ± .07 | −0.01 ± .04 | 3.3 ± .10 |
| 2000 | .00 ± .04 | .05 ± .02 | .08 ± .03 | .16 ± .03 | .43 ± .03 | .00 ± .04 | 3.3 ± .08 |

Table 56: Convergence of bias for benchmark methods, with **step** link and $\sigma_x = 0.1$

### C.3 Results with varying confounder stength

We present some additional results here experimenting with varying the strength of relationship between $X$ and $Z$. We do this by varying the $\sigma_X$ parameter in the data generating process described in Appendix B.1. Lower values of $\sigma_X$ mean that more of the variance of $X$ is explained by $Z$, corresponding to less confounding, whereas higher values of $\sigma_X$ indicate that less of the variance of $X$ is explained by $Z$, corresponding to more confounding.

We experimented with $\sigma_X$ values in the range $[0.1, 1.0, 4.0, 10.0]$, noting that $\sigma_X = 4.0$ is what was used in prior experiments, and we again used the **step** link function in this experiment. Our results are presented below. As expected as the level of confounding increases, the performance of the benchmark methods strongly decrease. However, our methods appear to be robust to the increasing level of confounding, maintaining close to zero bias with sufficiently low levels of regularization.

| n | $\textbf{OptZ}_{0.001}$ | $\textbf{OptZ}_{0.2}$ | $\textbf{OptZ}_{1.0}$ | $\textbf{OptZ}_{5.0}$ |
|---|---|---|---|---|
| 200 | $.33 \pm .05$ | $.23 \pm .02$ | $.30 \pm .03$ | $.74 \pm .03$ |
| 500 | $.13 \pm .01$ | $.12 \pm .01$ | $.16 \pm .01$ | $.37 \pm .02$ |
| 1000 | $.13 \pm .02$ | $.06 \pm .01$ | $.11 \pm .01$ | $.24 \pm .01$ |
| 2000 | $.06 \pm .01$ | $.04 \pm .00$ | $.09 \pm .01$ | $.16 \pm .01$ |

Table 57: Convergence of RMSE for weighted estimator using our weights, with **step** link and $\sigma_x = 1.0$

| n | **IPS** | $\textbf{OptX}_{0.001}$ | $\textbf{OptX}_{0.2}$ | $\textbf{OptX}_{1.0}$ | $\textbf{OptX}_{5.0}$ | **DirX** | **DirZ** |
|---|---|---|---|---|---|---|---|
| 200 | $.20 \pm .03$ | $2.2 \pm .04$ | $2.4 \pm .04$ | $2.7 \pm .04$ | $3.1 \pm .04$ | $.41 \pm .05$ | $3.1 \pm .05$ |
| 500 | $.12 \pm .01$ | $2.1 \pm .03$ | $2.3 \pm .03$ | $2.6 \pm .03$ | $3.0 \pm .03$ | $.22 \pm .02$ | $3.1 \pm .03$ |
| 1000 | $.08 \pm .01$ | $2.0 \pm .02$ | $2.1 \pm .02$ | $2.5 \pm .01$ | $3.0 \pm .01$ | $.09 \pm .01$ | $3.2 \pm .02$ |
| 2000 | $.05 \pm .01$ | $1.7 \pm .01$ | $1.9 \pm .01$ | $2.3 \pm .01$ | $2.8 \pm .01$ | $.06 \pm .01$ | $3.2 \pm .01$ |

Table 58: Convergence of RMSE for benchmark methods, with **step** link and $\sigma_x = 1.0$

| n | $\textbf{OptZ}_{0.001}$ | $\textbf{OptZ}_{0.2}$ | $\textbf{OptZ}_{1.0}$ | $\textbf{OptZ}_{5.0}$ |
|---|---|---|---|---|
| 200 | $-0.15 \pm .29$ | $.17 \pm .15$ | $.26 \pm .15$ | $.72 \pm .16$ |
| 500 | $.07 \pm .11$ | $.08 \pm .09$ | $.14 \pm .08$ | $.36 \pm .09$ |
| 1000 | $.07 \pm .10$ | $.04 \pm .05$ | $.10 \pm .05$ | $.23 \pm .04$ |
| 2000 | $.02 \pm .05$ | $.02 \pm .04$ | $.08 \pm .04$ | $.16 \pm .03$ |

Table 59: Convergence of bias for weighted estimator using our weights, with **step** link and $\sigma_x = 1.0$

| n | **IPS** | $\textbf{OptX}_{0.001}$ | $\textbf{OptX}_{0.2}$ | $\textbf{OptX}_{1.0}$ | $\textbf{OptX}_{5.0}$ | **DirX** | **DirZ** |
|---|---|---|---|---|---|---|---|
| 200 | $.07 \pm .18$ | $2.2 \pm .24$ | $2.4 \pm .23$ | $2.7 \pm .22$ | $3.1 \pm .21$ | $.21 \pm .35$ | $3.1 \pm .30$ |
| 500 | $.04 \pm .11$ | $2.1 \pm .18$ | $2.3 \pm .18$ | $2.6 \pm .17$ | $3.0 \pm .16$ | $.09 \pm .20$ | $3.1 \pm .19$ |
| 1000 | $.04 \pm .07$ | $1.9 \pm .09$ | $2.1 \pm .08$ | $2.5 \pm .08$ | $3.0 \pm .07$ | $.04 \pm .08$ | $3.2 \pm .10$ |
| 2000 | $.03 \pm .05$ | $1.7 \pm .08$ | $1.9 \pm .07$ | $2.3 \pm .07$ | $2.8 \pm .07$ | $.05 \pm .04$ | $3.2 \pm .06$ |

Table 60: Convergence of bias for benchmark methods, with **step** link and $\sigma_x = 1.0$

| n | $\textbf{OptZ}_{0.001}$ | $\textbf{OptZ}_{0.2}$ | $\textbf{OptZ}_{1.0}$ | $\textbf{OptZ}_{5.0}$ |
|---|---|---|---|---|
| 200 | $.39 \pm .07$ | $.24 \pm .02$ | $.36 \pm .02$ | $.81 \pm .02$ |
| 500 | $.19 \pm .02$ | $.18 \pm .02$ | $.23 \pm .02$ | $.49 \pm .02$ |
| 1000 | $.11 \pm .01$ | $.11 \pm .01$ | $.13 \pm .01$ | $.27 \pm .01$ |
| 2000 | $.08 \pm .01$ | $.08 \pm .01$ | $.09 \pm .01$ | $.17 \pm .01$ |

Table 61: Convergence of RMSE for weighted estimator using our weights, with **step** link and $\sigma_x = 4.0$

| n | **IPS** | $\textbf{OptX}_{0.001}$ | $\textbf{OptX}_{0.2}$ | $\textbf{OptX}_{1.0}$ | $\textbf{OptX}_{5.0}$ | **DirX** | **DirZ** |
|---|---|---|---|---|---|---|---|
| 200 | $.47 \pm .03$ | $2.0 \pm .03$ | $2.1 \pm .03$ | $2.3 \pm .02$ | $2.5 \pm .02$ | $.52 \pm .02$ | $2.6 \pm .02$ |
| 500 | $.48 \pm .03$ | $2.0 \pm .02$ | $2.1 \pm .02$ | $2.3 \pm .02$ | $2.6 \pm .02$ | $.48 \pm .02$ | $2.6 \pm .01$ |
| 1000 | $.39 \pm .02$ | $2.0 \pm .01$ | $2.1 \pm .01$ | $2.3 \pm .01$ | $2.5 \pm .01$ | $.48 \pm .02$ | $2.6 \pm .01$ |
| 2000 | $.40 \pm .01$ | $2.0 \pm .01$ | $2.1 \pm .01$ | $2.3 \pm .01$ | $2.5 \pm .01$ | $.45 \pm .02$ | $2.6 \pm .01$ |

Table 62: Convergence of RMSE for benchmark methods, with **step** link and $\sigma_x = 4.0$

| n | $\textbf{OptZ}_{0.001}$ | $\textbf{OptZ}_{0.2}$ | $\textbf{OptZ}_{1.0}$ | $\textbf{OptZ}_{5.0}$ |
|---|---|---|---|---|
| 200 | $.03 \pm .39$ | $.11 \pm .21$ | $.29 \pm .21$ | $.78 \pm .18$ |
| 500 | $.09 \pm .17$ | $.10 \pm .15$ | $.17 \pm .16$ | $.47 \pm .15$ |
| 1000 | $.02 \pm .11$ | $.05 \pm .09$ | $.08 \pm .09$ | $.25 \pm .09$ |
| 2000 | $.03 \pm .07$ | $.05 \pm .06$ | $.07 \pm .07$ | $.16 \pm .07$ |

Table 63: Convergence of bias for weighted estimator using our weights, with **step** link and $\sigma_x = 4.0$

| n | **IPS** | $\textbf{OptX}_{0.001}$ | $\textbf{OptX}_{0.2}$ | $\textbf{OptX}_{1.0}$ | $\textbf{OptX}_{5.0}$ | **DirX** | **DirZ** |
|---|---|---|---|---|---|---|---|
| 200 | $.40 \pm .25$ | $1.9 \pm .21$ | $2.1 \pm .20$ | $2.3 \pm .19$ | $2.5 \pm .18$ | $.49 \pm .18$ | $2.6 \pm .14$ |
| 500 | $.43 \pm .21$ | $2.0 \pm .16$ | $2.1 \pm .15$ | $2.3 \pm .14$ | $2.6 \pm .13$ | $.45 \pm .16$ | $2.6 \pm .12$ |
| 1000 | $.37 \pm .12$ | $2.0 \pm .10$ | $2.1 \pm .09$ | $2.3 \pm .09$ | $2.5 \pm .08$ | $.46 \pm .15$ | $2.6 \pm .11$ |
| 2000 | $.39 \pm .10$ | $2.0 \pm .08$ | $2.1 \pm .07$ | $2.3 \pm .07$ | $2.5 \pm .07$ | $.42 \pm .17$ | $2.6 \pm .11$ |

Table 64: Convergence of bias for benchmark methods, with **step** link and $\sigma_x = 4.0$

| n | $\mathbf{OptZ}_{0.001}$ | $\mathbf{OptZ}_{0.2}$ | $\mathbf{OptZ}_{1.0}$ | $\mathbf{OptZ}_{5.0}$ |
|---|---|---|---|---|
| 200 | $.60 \pm .07$ | $.37 \pm .04$ | $.53 \pm .04$ | $.97 \pm .03$ |
| 500 | $.42 \pm .05$ | $.32 \pm .04$ | $.39 \pm .03$ | $.69 \pm .03$ |
| 1000 | $.27 \pm .03$ | $.27 \pm .03$ | $.32 \pm .03$ | $.52 \pm .02$ |
| 2000 | $.20 \pm .03$ | $.18 \pm .02$ | $.24 \pm .02$ | $.40 \pm .02$ |

Table 65: Convergence of RMSE for weighted estimator using our weights, with **step** link and $\sigma_x = 10.0$

| n | **IPS** | $\mathbf{OptX}_{0.001}$ | $\mathbf{OptX}_{0.2}$ | $\mathbf{OptX}_{1.0}$ | $\mathbf{OptX}_{5.0}$ | **DirX** | **DirZ** |
|---|---|---|---|---|---|---|---|
| 200 | $.95 \pm .05$ | $1.6 \pm .04$ | $1.6 \pm .04$ | $1.7 \pm .04$ | $1.8 \pm .04$ | $1.2 \pm .06$ | $1.8 \pm .04$ |
| 500 | $.91 \pm .03$ | $1.6 \pm .02$ | $1.6 \pm .02$ | $1.7 \pm .02$ | $1.9 \pm .02$ | $.96 \pm .05$ | $1.8 \pm .03$ |
| 1000 | $.91 \pm .02$ | $1.6 \pm .02$ | $1.6 \pm .01$ | $1.7 \pm .01$ | $1.9 \pm .01$ | $.92 \pm .03$ | $1.9 \pm .02$ |
| 2000 | $.92 \pm .02$ | $1.6 \pm .02$ | $1.6 \pm .01$ | $1.7 \pm .01$ | $1.9 \pm .01$ | $.92 \pm .02$ | $1.9 \pm .02$ |

Table 66: Convergence of RMSE for benchmark methods, with **step** link and $\sigma_x = 10.0$

| n | $\mathbf{OptZ}_{0.001}$ | $\mathbf{OptZ}_{0.2}$ | $\mathbf{OptZ}_{1.0}$ | $\mathbf{OptZ}_{5.0}$ |
|---|---|---|---|---|
| 200 | $.01 \pm .60$ | $.23 \pm .29$ | $.48 \pm .23$ | $.95 \pm .18$ |
| 500 | $.01 \pm .42$ | $.19 \pm .26$ | $.34 \pm .20$ | $.68 \pm .14$ |
| 1000 | $.06 \pm .27$ | $.13 \pm .24$ | $.25 \pm .21$ | $.50 \pm .15$ |
| 2000 | $.03 \pm .20$ | $.10 \pm .15$ | $.19 \pm .14$ | $.38 \pm .13$ |

Table 67: Convergence of bias for weighted estimator using our weights, with **step** link and $\sigma_x = 10.0$

| n | **IPS** | $\mathbf{OptX}_{0.001}$ | $\mathbf{OptX}_{0.2}$ | $\mathbf{OptX}_{1.0}$ | $\mathbf{OptX}_{5.0}$ | **DirX** | **DirZ** |
|---|---|---|---|---|---|---|---|
| 200 | $.92 \pm .26$ | $1.5 \pm .23$ | $1.6 \pm .22$ | $1.7 \pm .22$ | $1.8 \pm .21$ | $1.1 \pm .35$ | $1.8 \pm .24$ |
| 500 | $.89 \pm .19$ | $1.6 \pm .13$ | $1.6 \pm .13$ | $1.7 \pm .12$ | $1.9 \pm .12$ | $.92 \pm .26$ | $1.8 \pm .19$ |
| 1000 | $.91 \pm .12$ | $1.6 \pm .08$ | $1.6 \pm .08$ | $1.7 \pm .08$ | $1.9 \pm .08$ | $.91 \pm .18$ | $1.9 \pm .12$ |
| 2000 | $.92 \pm .11$ | $1.6 \pm .08$ | $1.6 \pm .08$ | $1.7 \pm .08$ | $1.9 \pm .08$ | $.91 \pm .11$ | $1.9 \pm .11$ |

Table 68: Convergence of bias for benchmark methods, with **step** link and $\sigma_x = 10.0$

## Footnotes

[3]https://pypi.org/project/quadprog/