[Reviews · NeurIPS 2019]

Reviewer 1



# Originality The problem, i.e., policy evaluation in presence of unobserved confounders, is not quite new and there exist a great deal of previous works on this topic. But it is worth noting that this paper provided some rigoriously theoretical results, albeit with some strong assumptions. # Quality The theoretical part of this paper is technically sound but the experimental part appears not that convicing. The setup seems a bit simple. I think it is necessary to conduct one or two more complicated or real-world experiments, say, higher-dimensional Z? more complicated distributions over Z? more complex function of Y? etc. # clarity The paper is organised well and is easy to follow. The notations are also presented clearly. # Significance Although the results are kind of interest, they are quite limited in real world due to strong assumptions. For example, the authors assume access to an identified latent confounder model, which is almost impossible in practice. Also, some assumptions in Section 3.3 are not likely to occur in real world applications. As such, the proposed approach is probably of limited importance in practice.

Reviewer 2



Update: I am happy with the author's response. Thus I will keep my original score. Original comments: In this paper, the authors proposed a new method for policy evaluation when one only has proxies for true confounders. In particular, the author shows that the policy value can be consistently estimated in square root-n convergence rate when the outcome function is unknown. Pros. Policy evaluation with the existence of latent confounders is a very important question in contextual bandits. Although there is a large body of literature when assuming unconfoundedness, the problem of developing efficient policy evaluation method when allowing for latent confounders has been seldomly investigated. It’s good to see that this paper has filled this gap. Cons. In this paper, the author only provides explicit bounds and algorithms when the function class follows a RKHS. It would be interesting to see explicit bound and implementation of such algorithm in other class of models, such as neural networks.

Reviewer 3



This paper aims to evaluate a policy when the treatment and the outcome are confounded by some unobserved variables. In general, this problem is impossible and the authors assume that there are some proxy variables for the hidden confounders. The authors propose to solve the policy evaluation problem by designing a weighted estimator which computes a weight for an instance (outcome, treatment, proxy triple). When there is no unobserved confounder, the inverse propensity weights make the weighted estimator unbiased irrespective of the outcome. The authors first show that, in the presence of proxy variables, there exist weights that achieve an unbiased evaluation of the policy. However, such weights depend on the mean outcome functions and cannot be computed without knowledge of the outcome function. This existence of such weights motivates the authors to consider the minimization of conditional mean squared error over a class of functions. If the weights provide an error of O(1/n) for the CMSE and the outcome function belongs to the class of functions considered, then the resulting weighted estimator will be consistent. In order to handle the CMSE, the paper then considers a suitable upper bound with a loss of at most O(1/n) and try to minimize the upper bound. The upper bound has many interesting properties, in particular, it becomes a quadratic function of the weights when the class of functions is in RKHS. This immediately gives a QP to derive the weights when the mean outcome functions come from RKHS and consistency follows. Subsequently, the authors perform a simulation study and compare their method with various other policy evaluation methods. The proposed method seems to converge with an increasing number of samples, whereas methods like inverse propensity score or direct substitution do not. I found the paper provides a significant contribution to the literature on policy evaluation. The general framework of minimizing an adversarial balance objective might be useful for other settings like instrumental variables. I thought the paper was well-written, the motivation behind the theorems and the lemmas were clear and I could certainly follow the arguments. I have a couple of questions and some minor suggestions for the authors. 1. Following theorem 4, the text mentions matrix \Sigma. I don’t think such a matrix was introduced in the text. 2. For the simulation setting, the proxy variables have dimension 10 and the latent variables have dimension 1. Do the authors hope to obtain similar results for high-dimensional latent variables? 3. The proxy variables can also be weakly influenced by the latent confounders. It would be nice to see how the proposed methods are robust to the strength of the influence of the unobserved confounders to the proxy variables. Even a discussion in this regard will be helpful for the readers. 4. I am a bit surprised to see that there is no edge from X to T. Although the outcomes are completely determined by the treatment and the hidden confounder (say intelligence), the proxies (e.g. test scores) might affect the treatment assignment. Can it be incorporated into the current framework?

[Author Response · NeurIPS 2019]

We would like to thank the reviewers for their useful, detailed feedback! We will update the paper with the suggested minor revisions regarding typos and presentation improvements, and respond to individual reviewers comments below.

Reviewer #1:

1. "Originality": we are aware of no paper looking at policy evaluation in the setting we consider where we only have proxies for the confounders, but the latent model is identifiable. We cite the relevant papers on effect estimation from proxies and on policy evaluation in unidentifiable settings.

2. "Quality": In order to actually evaluate proxy methods out-of-sample, data *has to be simulated*. The supplement already includes results from a range of link functions ("more complex function of Y"). Changing the dimensions of $X, Z$ gave similar comparative performance (see R4 pt2 for more detail). We can also add the Twins dataset from Louizos et al. [26] to the supplement but it is also simulated (simulated treatment assignment $T$ and proxies $X$).

3. "Significance":

    (a) Regarding the assumption of an identified latent confounder model: we reference the rich literature on the conditions and methods for identifying such a model, and we focus on the downstream task of policy evaluation, where there is a lack of previous literature. Therefore our research should be seen as complementary to this existing work.

    (b) It is true that, in finite samples, an estimated latent model will have errors, but our algorithm uses this model only to approximate the $Q$ matrix and this error can be made to vanish as $n \to \infty$. In particular, Thm 3 (convergence) trivially holds also if we use any $\hat{\phi} \to \phi$ in $L_1$ in probability by Slutsky's Theorem (we will update the theorem and proof accordingly).

    (c) All of the assumptions in Section 3.3 are assumptions about the choice of function class $\mathcal{F}$, not about the data distribution, and we show they are specified by particular choices (RKHS), so they absolutely do not limit the practical applicability of our method as none of those assumptions has to be tested/verified. One substantial assumption is specification ($\mu \in \mathcal{F}$), which is common and necessary for consistency. We can make the method nonparametric and use a universal kernel (e.g., RBF) to avoid this; a trivial corollary to Thm 3 using universal approximators (such as RBF RKHS) would give slower but specification-free $o_p(1)$ convergence (because we can get an error bound of $\epsilon + O_p(1/\sqrt{n})$ for any $\epsilon > 0$ using Thm 3 and universal $L_\infty$ approximation of $\mu$, where the $O_p$ term's constant can depend on $\epsilon$).

Reviewer #3:

1. "Cons": Our convergence theory in Sec 3 is general and easily extends easily to neural nets with weight decay, as they immediately satisfy all of the assumptions (we will update to make this more explicit). Regarding a concrete implementation of such a NN-based algorithm, that would be very involved (optimizing a GAN-like adversarial neural net objective, which is known to be challenging), and is far beyond the scope of our paper, but we think it would be a promising direction for future work.

2. "Improvements": See R1 pt2, R4 pt2. In order to actually evaluate proxy methods out-of-sample, data *has to be simulated*. We can add results from the Twins dataset from Louizos et al. [26] to the supplement with the other extra results but note that it is also simulated.

Reviewer #4:

1. Thanks for catching. This is any positive definite diagonal covariance matrix. We will update paper to define this.

2. Our pilot experiments with different dimensions of $X$ and $Z$ indicated similar comparative performance. We chose the dimensions for our experiments for reasons of convenience (as this setup gave us low sample complexity and allowed running many replications of interesting experiments on fewer data points). We will provide some additional numbers in the supplement. The code (on GitHub) allows anyone to try and tinker with higher dimensions. (Note there is already a plethora of additional results in supplemental Sec C.)

3. We like your suggestion on exploring the effect of the strength of relationship between $Z$ and $X$, and are going to run an additional experiment (to place in supplement) to explore this effect. Concretely, we can measure strength of relationship by the fraction of $X$ variance explained by $Z$, so we can just vary $\sigma_X$ on line 245 and run the same experiment over an array of different relationship strengths.

4. You are correct. In our theory the only part of Assumption 1 that we actually use is that for every $t$, $Y(t)$ is conditionally independent of $(X, T)$, given $Z$, so there is nothing prohibiting an arrow from $X$ to $T$ in the DAG in Figure 1. We will update Assumption 1 and Figure 1 accordingly. (Note that all our math stays exactly the same. All that will change is adding $X \to T$ in Figure 1 and rephrasing Assumption 1.)

[Meta-Review · NeurIPS 2019]

This paper makes some significant contributions to the problem of policy evaluation in the presence of unobserved confounders (assuming access to some proxy variables). The theoretical and technical contributions of this paper are pretty significant. There have been some inputs from reviewers regarding improving the experimental setup of this paper by utilizing more real-world high dimensional data. Based on reviewer feedback and rebuttals, I recommend this paper to be accepted.